# TCA-DiT: Quantizing Diffusion Transformers via Temporal Channel Alignment

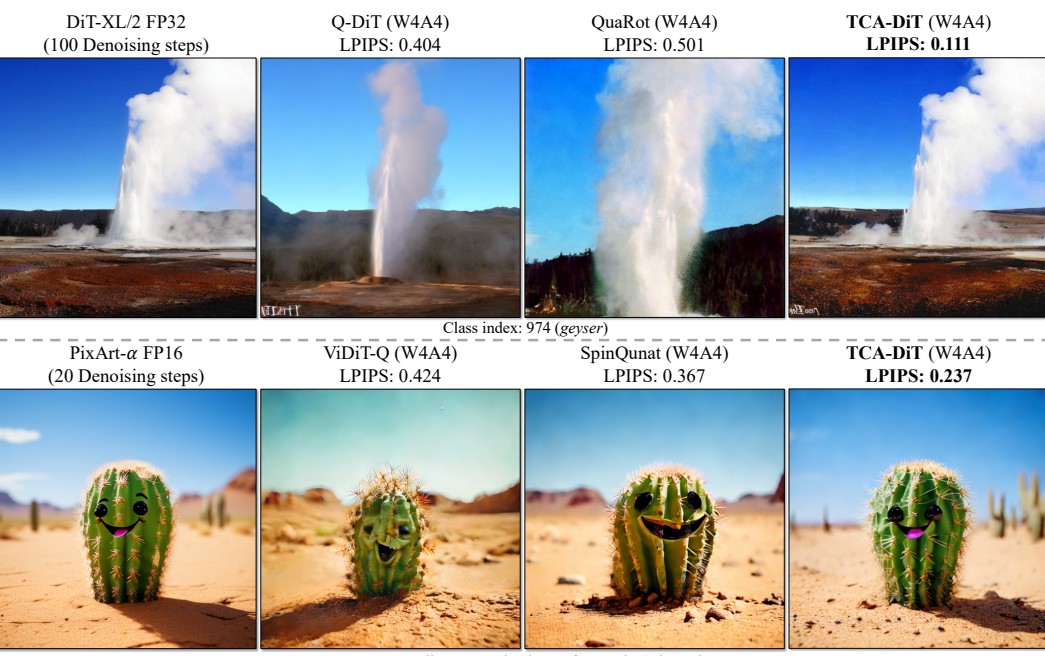

Figure 1: TCA-DiT is a post-training quantization framework for Diffusion Transformers, supporting 4-bit weights and activations while effectively preserving visual fidelity.

## Abstract

Diffusion Transformers (DiTs) have achieved remarkable success in generative modeling, but their deployment is hindered by massive model sizes and high inference costs. Post-Training Quantization (PTQ) offers a retraining-free compression paradigm, yet its application to DiTs is particularly challenging due to timestep-varying, channel-wise activation anomalies. These anomalies vary dynamically across timesteps, undermining existing rotation- or scaling-based PTQ methods and leaving residual misaligned anomaly channels that impair quantization fidelity. We propose **TCA-DiT**—**T**emporal **C**hannel **A**lignment for **Di**ffusion **T**ransformers—a PTQ framework designed to explicitly address such timestep-varying anomalies. Specifically, we first introduce *Anomaly-aware Rotation Calibration (ARC)*, a learnable rotation-scaling mechanism that jointly optimizes rotation matrices with reconstruction and anomaly alignment losses, thereby aligning anomaly channels across timesteps and enabling more precise per-channel scaling. To improve calibration efficiency, we further develop *Anomaly-guided Timestep Grouping (ATG)*, which clusters timesteps based on anomaly distributions, capturing full temporal dynamics with a compact set of representatives. Finally, we propose *Reordered Group Quantization (RGQ)*, which reorders channels before group quantization to reduce intra-group variance and minimize quantization error. On DiT-XL/2 with W4A4, TCA-DiT improves FID by **0.74** and **6.47** on ImageNet 256×256 and 512×512, respectively. On PixArt-$\alpha$, it achieves a substantial **3.74** FID improvement while reducing memory usage by **3.8×** and accelerating inference by **3.5×**. These results highlight the critical role of anomaly alignment in enabling both effective and efficient quantization of DiTs.

# 1 INTRODUCTION

Diffusion Transformers (DiTs) Peebles & Xie (2023) have recently emerged as powerful backbones for generative modeling, achieving SOTA performance in high-resolution image synthesis—e.g., PixArt-$\alpha$ Chen et al. (2024)—and supporting large-scale video generation systems such as OpenAI's Sora Brooks et al. (2024). Despite these successes, DiTs are characterized by massive parameter counts and iterative denoising, which result in high latency and memory overhead. Such limitations highlight the need for effective compression and acceleration techniques for practical deployment.

Model quantization Jacob et al. (2018a); Nagel et al. (2021) is a widely adopted approach to reducing memory footprint and computational cost. Among the various techniques, Post-Training Quantization (PTQ) Nagel et al. (2020); Li et al. (2021) is particularly appealing, as it enables low-bit quantization of both weights and activations without retraining. This property makes PTQ especially suitable for DiTs, where full fine-tuning is computationally prohibitive.

However, applying PTQ to DiTs poses unique challenges. While **activation outliers** are well-known obstacles, DiTs exacerbate the issue. For instance, in the `blocks.0.attn.qkv` layer of DiT-XL/2 under a 100-step DDPM Ho et al. (2020) scheduler, certain channels generate extreme values at different timesteps ($t = 99, 50, 0$), resulting in highly skewed distributions (Fig. 2a). These rare yet large activations expand the dynamic range, forcing uniform quantization to allocate precision to a few outliers while compressing the majority, thereby introducing substantial quantization errors.

A key challenge in DiTs is the temporal shift of anomaly channels, which leads to instability across timesteps. As shown by substantial KL divergences between $t = 50$ and $t = 99, 0$ (Fig. 2a), DiTs exhibit large discrepancies across timesteps, undermining suppression methods that assume static patterns. Scaling-

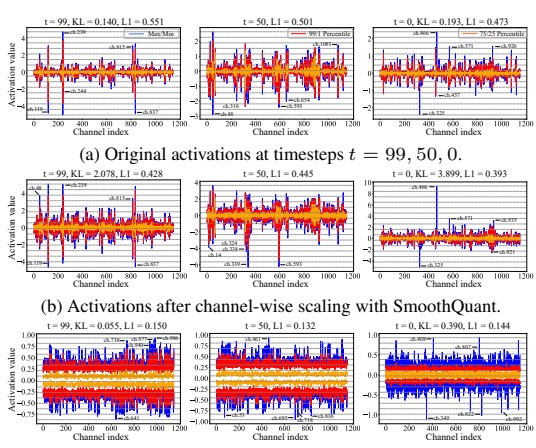

(a) Original activations at timesteps $t = 99, 50, 0$.

(b) Activations after channel-wise scaling with SmoothQuant.

(c) Activations after fixed Hadamard rotation with QuaRot.

Figure 2: Channel-wise activation distributions from `blocks.0.attn.qkv` in DiT-XL/2 across timesteps. Relative L1 quantization error (4-bit per-token) and KL divergence (w.r.t. $t = 50$) are reported.

based approaches such as SmoothQuant Xiao et al. (2023) provide limited benefits when applied with timestep-averaged statistics (Fig. 2b): averaging dilutes timestep-specific anomalies, while global maxima over-suppress mild activations. Timestep-specific scaling could mitigate this issue, but it requires multiple weight versions and complicates deployment. Rotation-based methods (e.g., Hadamard transforms Ashkboos et al. (2024)) suffer from a similar limitation: a single random rotation cannot adapt to timestep-varying distributions, leaving residual outliers (Fig. 2c). Recent PTQ methods for DiTs inherit these drawbacks. For example, ViDiT-Q Zhao et al. (2025) combines scaling and rotation, while timestep-averaged scaling weakens suppression at highly abnormal steps, and a single random rotation fails to capture temporal variations. Similarly, DiTAS Dong & Zhang (2025) extends SmoothQuant through Temporal-averaged Smoothing, computing per-channel scaling from global maxima across timesteps. While robust to sporadic spikes, it over-suppresses channels at mild timesteps, resulting in unnecessary smoothing. Overall, existing methods share a common limitation: none explicitly align anomaly channels across timesteps. Motivated by this, we propose a **rotate-then-scale** strategy, where an **anomaly alignment loss** first guides rotation to synchronize anomaly channels across timesteps, after which scaling suppresses them more effectively, yielding smoother activation distributions and lower quantization error (Fig. 4c).

Another challenge stems from channel-wise heterogeneity. Even after rotation and scaling, channel distributions remain uneven. While group quantization Zhao et al. (2024); Lin et al. (2024) alleviates mismatch by localizing parameter estimation, fixed partitions often cluster incompatible channels, inflating intra-group variance and amplifying quantization error. To address this, Q-DiT Chen et al. (2025) employs fine-grained group quantization with an evolutionary algorithm to search for optimal group sizes per layer, directly guided by FID. Although effective, this strategy is prohibitively expensive, as each fitness evaluation requires generating hundreds of samples, making the search

extremely time-consuming. A pilot study on the `blocks.26.attn.proj` layer of DiT-XL/2 further reveals that default channel ordering is suboptimal: random permutations can reduce reconstruction error and strongly correlate with FID (Fig. 5b). This finding highlights the importance of channel ordering and the need for efficient search strategies in group quantization. Motivated by this, we leverage evolutionary search but adopt a lightweight layer-wise reconstruction loss as the objective, which correlates well with generation quality while avoiding costly sampling.

To address these challenges, we propose **TCA-DiT**—**T**emporal **C**hannel **A**lignment for **Di**ffusion **T**ransformers—a PTQ framework tailored to the temporal dynamics of DiTs (Fig. 3). The core idea is to align timestep-varying activation anomalies before scaling. Specifically, we introduce *Anomaly-aware Rotation Calibration (ARC)*, a learnable rotation-scaling mechanism designed to mitigate temporal instability. A shared rotation matrix is jointly optimized with two complementary losses: (i) a **layer reconstruction loss** to preserve fidelity across timesteps, and (ii) an **anomaly alignment loss** to enforce consistency of residual anomaly channels. This explicit alignment enables more accurate per-channel scaling, after which a diagonal scaling matrix suppresses the aligned anomalies. To reduce calibration cost, we propose *Anomaly-guided Timestep Grouping (ATG)*, which clusters timesteps with similar anomaly distributions. Unlike naive partitioning, ATG employs constrained hierarchical clustering based on KL divergence, ensuring that a compact set of representatives captures the full temporal diversity of activations. Finally, *Reordered Group Quantization (RGQ)* addresses residual heterogeneity after transformation by introducing a lightweight reordering step. Using evolutionary search to identify channel permutations that minimize reconstruction error, RGQ reduces intra-group variance and improves quantization fidelity. Visual examples in Fig. 1 demonstrate these improvements. Our main contributions are summarized as follows:

- We propose **TCA-DiT**, a temporal anomaly-aware PTQ framework that explicitly addresses timestep-varying anomalies and channel misalignment in DiTs.

- We introduce three complementary components: ARC for anomaly alignment across timesteps and precise scaling, ATG for efficient calibration via distribution-aware clustering, and RGQ for reducing intra-group variance through channel reordering.

- TCA-DiT achieves state-of-the-art PTQ performance on DiTs while remaining hardware-friendly and inference-efficient. On DiT-XL/2 and PixArt-$\alpha$, it consistently improves FID under low-bit settings while delivering up to **3.8**$\times$ memory savings and **3.5**$\times$ speedup, underscoring the importance of anomaly alignment for practical deployment.

## 2 RELATED WORK

### 2.1 TRANSFORMER QUANTIZATION

Model quantization compresses high-precision tensors (e.g., FP32) into low-bit integers (Nagel et al., 2021), thereby reducing memory footprint and accelerating inference (Jacob et al., 2018b). This work focuses on hardware-friendly **uniform quantization**, defined as:

$$\boldsymbol{x}_q = Q(\boldsymbol{x}; s, z, k) = s \cdot \text{clip}\left(\lfloor \tfrac{\boldsymbol{x}}{s} \rceil + z, 0, 2^k - 1\right) - z, \tag{1}$$

where $\boldsymbol{x}$ is the input tensor, $s$ is the step size, and $z$ is the zero-point. Quantization parameters are typically optimized by minimizing layer reconstruction error[1] (Nagel et al., 2020).

A central challenge in transformer quantization is **activation outliers**, which expand the dynamic range and exacerbate quantization error. Existing approaches fall into two categories: scaling-based and rotation-based. Scaling-based methods, such as SmoothQuant Xiao et al. (2023), suppress outliers by redistributing magnitudes between activations and weights through channel-wise scaling: $\boldsymbol{Y} = \left(\boldsymbol{X}\text{diag}(\boldsymbol{s})^{-1}\right) \cdot \left(\text{diag}(\boldsymbol{s})\boldsymbol{W}\right)$, where $\boldsymbol{s}_i = \max(|\boldsymbol{X}_{:,i}|)^{\alpha}/\max(|\boldsymbol{W}_{:,i}|)^{1-\alpha}$ and $\alpha \in [0,1]$. OmniQuant Shao et al. (2024) jointly optimizes $\boldsymbol{s}$ with quantization parameters, while I-LLM Hu et al. (2024) adapts this approach for integer-only inference. Rotation-based methods instead decorrelate channels with an orthogonal transformation: $\boldsymbol{Y} = (\boldsymbol{X}\boldsymbol{R}) \cdot (\boldsymbol{R}^{\top}\boldsymbol{W})$, where $\boldsymbol{R}\boldsymbol{R}^{\top} = \boldsymbol{I}$, which redistributes activation energy more evenly. QuIP Chee et al. (2023) and QuaRot Ashkboos et al. (2024) employ random Hadamard transforms, while SpinQuant Liu et al. (2025) learns $\boldsymbol{R}$ by minimizing task loss. These methods are effective for LLMs, where outlier patterns are relatively stable.

---

[1]The derivation of why layer reconstruction loss serves as a good proxy for task loss is presented in Sec. A.

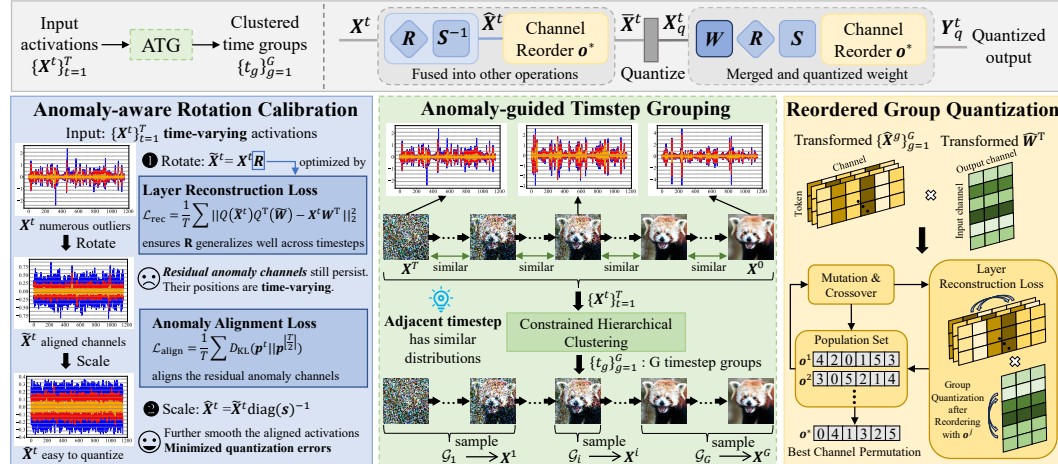

Figure 3: Overview of the proposed TCA-DiT framework.

In contrast, DiTs exhibit **timestep-varying anomalies**, where dominant channels shift as denoising progresses. This temporal variability renders static scaling factors and fixed rotations insufficient, highlighting adaptive mechanisms that explicitly account for shifting outlier distributions.

## 2.2 DiT Quantization

Recent works have explored PTQ strategies tailored to DiTs. PTQ4DiT Wu et al. (2024) introduces Channel-wise Salience Balancing to redistribute activation-weight magnitudes, and further proposes Spearman's $\rho$-guided Salience Calibration to emphasize highly abnormal timesteps. While effective, its fixed balance mask often oversmooths channels at less abnormal steps, reducing adaptability. Q-DiT Chen et al. (2025) combines group quantization with dynamic activation quantization. It employs evolutionary search to determine per-layer group sizes, reducing intra-group variance, and adapts quantization parameters at the sample level to follow timestep-varying activations. However, it does not explicitly suppress outliers. ViDiT-Q Zhao et al. (2025) proposes Static-Dynamic Channel Balancing, which first applies a timestep-averaged scaling factor to suppress *static* imbalance, followed by a random Hadamard rotation to smooth *dynamic* variations from timestep embeddings. While this mitigates extremes, suppression is diluted at highly abnormal steps, and a single random rotation is insufficient to address timestep-varying anomaly patterns. DiTAS Dong & Zhang (2025) introduces Temporal-aggregated Smoothing, deriving per-channel scaling factors from maximum activations across timesteps. This SmoothQuant extension improves robustness against sporadic spikes but often over-suppresses channels at mild timesteps, causing unnecessary smoothing. Overall, these methods reduce quantization degradation but share a key limitation: none explicitly align anomaly channels across timesteps. This motivates our **rotate-then-scale** framework, which first applies an **anomaly alignment loss** to guide rotations and synchronize residual anomaly channels, then suppresses them via scaling. This two-stage design enables more effective quantization of DiTs.

## 3 Method

### 3.1 Anomaly-aware Rotation Calibration

Consider a linear layer $f(\cdot; \boldsymbol{W})$ with $\boldsymbol{W} \in \mathbb{R}^{\mathcal{C}_o \times \mathcal{C}_i}$. At timestep $t$, the input $\boldsymbol{X}^t \in \mathbb{R}^{N \times \mathcal{C}_i}$ produces $\boldsymbol{Y}^t = \boldsymbol{X}^t \boldsymbol{W}^\top \in \mathbb{R}^{N \times \mathcal{C}_o}$, where $t = 1, \ldots, T$ indexes the denoising steps. As noted in Sec. 1, random Hadamard rotations redistribute activations but cannot adapt to timestep-varying outliers in DiTs. To address this, we propose Anomaly-aware Rotation Calibration (ARC), a two-stage *rotate-then-scale* framework. Unlike SpinQuant Liu et al. (2025), which learns task-specific rotations, ARC learns a *shared* rotation $\boldsymbol{R}$ per layer that generalizes across timesteps.

**Stage 1: Rotation with Temporal Alignment.** We first optimize $\boldsymbol{R}$ with a layer reconstruction loss averaged over the full denoising trajectory:

$$\mathcal{L}_{\text{rec}} = \frac{1}{T} \sum_{t=1}^{T} \|Q(\tilde{\boldsymbol{X}}^t) Q^\top(\tilde{\boldsymbol{W}}) - \boldsymbol{X}^t \boldsymbol{W}^\top\|_2^2, \tag{2}$$

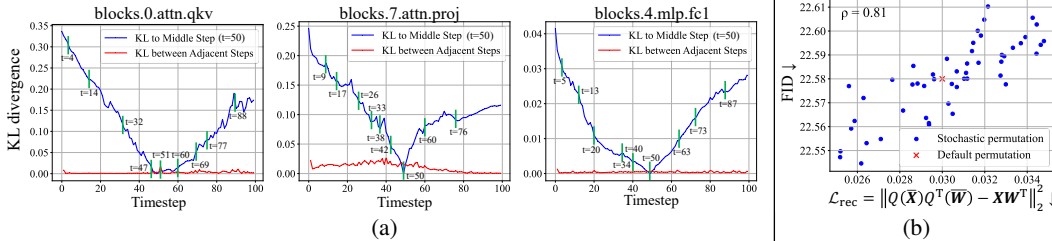

(a)                       (b)

Figure 5: (a) KL divergence of normalized activation distributions across timesteps in DiT-XL/2, measured against the midpoint and adjacent steps. Green lines mark partition boundaries selected by ATG. (b) Quantization performance vs. reconstruction loss under different channel permutations.

where $\tilde{X}^t = X^t R$, $\tilde{W} = WR$, and $Q(\cdot)$ is the quantization function from Eq. (1). While $\mathcal{L}_{\text{rec}}$ suppresses outliers, Fig. 4a shows that *residual anomaly channels* remain. For example, the five most dominant channels at timestep $t = 99, 50$, and $0$ show no overlap, indicating that different channels become anomalous at different stages. This inconsistency yields large KL divergences relative to the mid-timestep reference. Optimizing $R$ solely for reconstruction partially reduces anomaly magnitudes but fails to ensure temporal stability.

To explicitly stabilize anomalies, we introduce an **anomaly alignment loss** that enforces consistent channel dominance across timesteps:

$$\mathcal{L}_{\text{align}} = \frac{1}{T} \sum_{t=1,\ t\neq\lfloor\frac{T}{2}\rceil}^{T} D_{\text{KL}}(\boldsymbol{p}^t \| \boldsymbol{p}^{\lfloor\frac{T}{2}\rceil}), \qquad (3)$$

where $\boldsymbol{p}^t = \text{softmax}(\boldsymbol{a}^t)$, $a_i^t = \text{SW}_\gamma(|\tilde{X}_{:,i}^t|)$, $\text{SW}_\gamma(\boldsymbol{v}) = \sum_j v_j \text{softmax}(\gamma\boldsymbol{v})_j$, and $\gamma$ is a

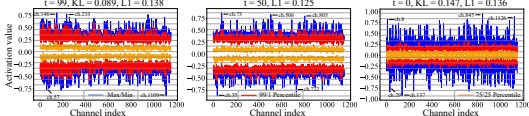

(a) Rotation with only **layer reconstruction loss**. Top-5 *residual anomaly channels* (see arrows) differ drastically across timesteps.

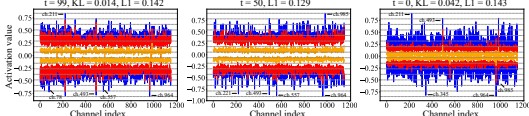

(b) Rotation with **anomaly alignment loss**. *Residual anomaly channels* become consistent across timesteps, reducing KL divergence.

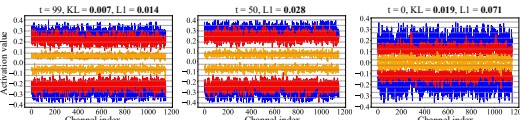

(c) **Full ARC**, combining rotation and scaling. Distributions are smoothed, and quantization error is further reduced.

Figure 4: Channel-wise activation distributions at different stages of Anomaly-aware Rotation Calibration, using the same setup as Fig. 2.

large scaling factor. We adopt the mid-timestep ($t = \lfloor T/2 \rceil$) as the anchor, balancing noisy early steps and data-dominated late steps. Early or late anchors generally increase variance due to unstable anomaly patterns. As illustrated in Fig. 4b, incorporating $\mathcal{L}_{\text{align}}$ aligns anomaly channels across timesteps, reducing KL divergence and improving stability. The joint objective is:

$$\min_{R} \mathcal{L} = \mathcal{L}_{\text{rec}} + \beta \cdot \mathcal{L}_{\text{align}}, \qquad (4)$$

where $\beta$ balances reconstruction fidelity and temporal alignment. This *soft alignment* ensures anomaly consistency across the trajectory, enabling more effective and precise scaling.

**Stage 2: Per-channel Scaling.** With anomaly channels aligned, ARC applies per-channel scaling:

$$\hat{Y}^t = \hat{X}^t \hat{W}^\top = \left(\tilde{X}^t \text{diag}(\boldsymbol{s})^{-1}\right)\left(\tilde{W}\text{diag}(\boldsymbol{s})\right)^\top; \quad s_i = \max_{j,t}\left(|\tilde{X}_{j,i}^t|\right)^\alpha / \max_j\left(|\tilde{W}_{j,i}|\right)^{1-\alpha}, \quad (5)$$

where $\alpha \in [0, 1]$ (0.5 by default). With temporally consistent anomalies, scaling factors can be estimated more reliably, yielding smoother distributions (Fig. 4c) and lower quantization error. Both $R$ and $\text{diag}(\boldsymbol{s})$ are fixed and can be fused into adjacent operators with negligible runtime overhead.

### 3.2 Anomaly-guided Timestep Grouping

As discussed in Sec. 3.1, both $\mathcal{L}_{\text{rec}}$ and $\mathcal{L}_{\text{align}}$ require activations from all $T$ timesteps, which becomes costly for long diffusion trajectories. Prior studies show that neighboring steps share similar distributions (Park et al., 2024; Teng et al., 2025). To verify, we compute per-channel distributions $\boldsymbol{q}^t = \text{softmax}(\boldsymbol{c}^t)$ with $c_i^t = \max(|X_{:,i}^t|)$, and evaluate $D_{\text{KL}}(\boldsymbol{q}^t \| \boldsymbol{q}^{t+1})$ between adjacent steps and $D_{\text{KL}}(\boldsymbol{q}^t \| \boldsymbol{q}^{\frac{T}{2}})$ relative to the midpoint. As shown in Fig. 5a, adjacent steps are nearly indistinguishable, while divergence grows smoothly from the midpoint—indicating gradual yet non-uniform evolution. Thus, naive uniform subsampling (e.g, every $k$ steps) risks overlooking critical transitions, leading to suboptimal calibration.

---

**Algorithm 1** Constrained Hierarchical Clustering for Timestep Grouping

---

**Require:** $T$ normalized distributions $\{\boldsymbol{q}^t\}_{t=1}^T$; target group number $G$.
 1: Initialize $T$ singleton groups, each containing one $\boldsymbol{q}^t$.
 2: **while** current number of groups $> G$ **do**
 3:     Compute the paired KL divergence $D_{\text{KL}_{i,j}}(\text{Group}_i, \text{Group}_j)$ between all adjacent groups ($|i-j|=1$).
 4:     Merge the adjacent pair with minimum $\mathcal{D}_{\text{KL}_{i,j}}$ into one group.
 5: **return** Partition boundaries $\{\tau_1, \tau_2, \cdots, \tau_{G-1}\}$.

---

To reduce cost while maintaining representativeness, we propose Anomaly-guided Timestep Grouping (ATG). ATG partitions the $T$ timesteps into $G$ *consecutive* groups of similar distributions ($G \ll T$), formulated as a constrained hierarchical clustering problem over $\{\boldsymbol{q}^t\}_{t=1}^T$ (Jarman, 2020):

$$\underset{\tau_1, \tau_2, \cdots, \tau_{G-1}}{\arg\min} \sum_{g=1}^{G} \sum_{t=1}^{T} \mathbf{1}_{\mathcal{G}(t)=g} D_{\text{KL}}(\boldsymbol{q}^t \| \boldsymbol{q}^g),$$

$$\textbf{s.t.} \begin{cases} 0 < \tau_1 < \tau_2 < \cdots < \tau_{G-1} < T, \\ \tau_{g-1} < t \leq \tau_g \text{ for } \forall \mathcal{G}(t) = g, \end{cases}$$

(6)

where $\boldsymbol{q}^g$ denotes the centroid distribution of group $g$. This can be solved efficiently using Alg. 1, with the average-linkage method Seifoddini (1989) for computing the paired KL divergence. As illustrated in Fig. 5a, $T = 100$ steps can be reduced to $G = 10$ groups, each capturing a distinct denoising stage. ATG then randomly samples one timestep from each group, yielding a compact calibration set $\{\boldsymbol{X}^g\}_{g=1}^G$. This approach substantially reduces calibration overhead and optimization cost while ensuring full trajectory coverage, enabling efficient and distribution-aware calibration.

## 3.3 Reordered Group Quantization

Per-token or per-channel quantization is applied to each activation or weight row but suffers from cross-channel mismatch. Group quantization Lin et al. (2024); Zhao et al. (2024) alleviates this by dividing channels into groups of size $g$, within which localized quantization is applied:

$$\boldsymbol{Y}_{i,j}^q = \sum_{k=0}^{\mathcal{C}_i} \boldsymbol{X}_{i,k}^q \boldsymbol{W}_{k,j}^{q\top} = \sum_{u=0}^{\mathcal{C}_i/g-1} \sum_{v=0}^{g} Q_u(\boldsymbol{X}_{i,ug+v}) Q_u^\top(\boldsymbol{W}_{ug+v,j}).$$

(7)

However, as noted in Sec. 1, distributional mismatch persists even after rotation and scaling. Default ordering often clusters incompatible channels, resulting in high intra-group variance and suboptimal quantization (Fig. 5b). This motivates reordering channels into more compatible groups.

We propose Reordered Group Quantization (RGQ), which searches for an optimal channel permutation $\boldsymbol{o}^\star(\mathcal{C}_i)$ prior to group quantization. RGQ employs an evolutionary search that iteratively refines candidate permutations using a layer-wise reconstruction loss:

$$\mathcal{L}_{\text{rec}} = \frac{1}{T} \sum_{g=1}^{G} \| Q_u(\bar{\boldsymbol{X}}^g) Q_u^\top(\bar{\boldsymbol{W}}) - \boldsymbol{X}^g \boldsymbol{W}^\top \|_2^2,$$

(8)

where $\bar{\boldsymbol{X}}^g = \hat{\boldsymbol{X}}_{:,\boldsymbol{o}}^g$ and $\bar{\boldsymbol{W}} = \hat{\boldsymbol{W}}_{:,\boldsymbol{o}}$ are activations and weights reordered according to permutation $\boldsymbol{o}$, and $Q_u$ denotes the group-wise quantizer. Iterative evolution (see Alg. 2 at Sec. B.1) converges to permutations that minimize reconstruction error. Once the optimal ordering $\boldsymbol{o}^\star(\mathcal{C}_i)$ is obtained, both activations and weights are reordered before applying group quantization. This reduces intra-group variance and quantization error. Moreover, following RPTQ Yuan et al. (2023), the reordering operator can be statically fused into components such as LayerNorm, incurring no runtime overhead during inference. **The complete TCA-DiT pipeline is summarized in Alg. 3 at Sec. B.2.**

## 4 Experiments

### 4.1 Experimental Setup and Implementation Details

**Class-conditioned Image Generation.** We first evaluate TCA-DiT on class-conditional image generation using pre-trained DiT-XL/2 models Peebles & Xie (2023) trained on ImageNet $256\times256$ and

| Timesteps | Bit-width (W/A) | Method | FID ↓ | sFID ↓ | IS ↑ | Precision ↑ | |
|---|---|---|---|---|---|---|---|
| | 32/32 | FP | 5.64 | 18.95 | 261.88 | 0.8094 | 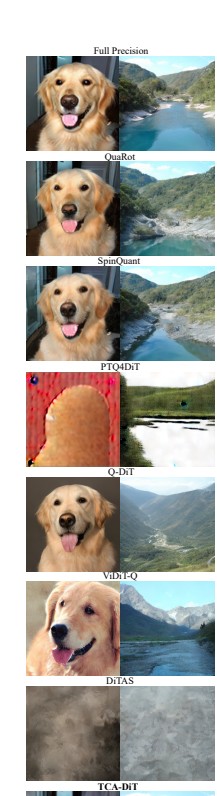 |
| 100 | 4/8 | SmoothQuant | 7.60 | 21.52 | 215.10 | 0.7413 | |
| | | QuaRot | 6.83 | 20.61 | 230.85 | 0.7646 | |
| | | SpinQuant | _6.74_ | 20.47 | _233.75_ | _0.7698_ | |
| | | PTQ4DiT | 9.24 | 23.65 | 179.75 | 0.7059 | |
| | | Q-DiT | 7.27 | 21.41 | 219.70 | 0.7505 | |
| | | ViDiT-Q | 9.65 | 23.18 | 191.90 | 0.6932 | |
| | | DiTAS | 7.06 | _20.36_ | 213.80 | 0.7626 | |
| | | **Ours** | **6.20** | **19.78** | **247.58** | **0.7922** | |
| | 4/4 | SmoothQuant | 26.17 | 83.98 | 101.55 | 0.5519 | |
| | | QuaRot | 9.96 | _24.61_ | 191.00 | _0.6988_ | |
| | | SpinQuant | _9.78_ | 24.90 | _192.37_ | 0.6862 | |
| | | PTQ4DiT | - | - | - | - | |
| | | Q-DiT | 28.72 | 82.81 | 93.20 | 0.5138 | |
| | | ViDiT-Q | 18.92 | 30.28 | 132.24 | 0.6137 | |
| | | DiTAS | - | - | - | - | |
| | | **Ours** | **9.04** | **24.01** | **203.05** | **0.7222** | |
| 50 | 32/32 | FP | 6.82 | 21.79 | 242.6 | 0.7792 | |
| | 4/8 | SmoothQuant | 10.79 | 25.84 | 182.50 | 0.7036 | |
| | | QuaRot | 9.22 | 24.63 | _199.69_ | _0.7340_ | |
| | | SpinQuant | 9.26 | 24.73 | 198.75 | 0.7225 | |
| | | PTQ4DiT | 10.70 | 24.51 | 164.18 | 0.6839 | |
| | | Q-DiT | 10.27 | 25.69 | 185.14 | 0.7109 | |
| | | ViDiT-Q | 14.17 | 28.28 | 159.45 | 0.6554 | |
| | | DiTAS | _9.20_ | _23.84_ | 191.48 | 0.7320 | |
| | | **Ours** | **7.54** | **23.07** | **223.13** | **0.7624** | |
| | 4/4 | SmoothQuant | 31.01 | 89.73 | 87.19 | 0.5129 | |
| | | QuaRot | 14.58 | _30.39_ | 161.81 | 0.6692 | |
| | | SpinQuant | _14.10_ | 30.47 | _164.85_ | _0.6714_ | |
| | | PTQ4DiT | - | - | - | - | |
| | | Q-DiT | 34.98 | 89.26 | 79.29 | 0.4705 | |
| | | ViDiT-Q | 27.43 | 38.12 | 106.41 | 0.5563 | |
| | | DiTAS | - | - | - | - | |
| | | **Ours** | **12.76** | **29.64** | **172.53** | **0.6871** | |

Figure 6: Performance of class-conditioned image generation on ImageNet 256×256. **Left:** Quantitative comparison of different PTQ methods across bit-widths and denoising steps (cfg=1.5). **Right:** Visual comparison of generated samples under W4A4 with 100 denoising steps (cfg=4.0).

512×512 resolutions. Sampling is conducted with the DDPM solver Ho et al. (2020) using 100 and 50 denoising steps, combined with Classifier-Free Guidance (CFG) Ho & Salimans (2022) at scale 1.5. Image quality is assessed with four standard metrics: *Fréchet Inception Distance* (FID) Heusel et al. (2017), *spatial FID* (sFID) Salimans et al. (2016), *Inception Score* (IS) Barratt & Sharma (2018), and *Precision*. Following convention, 10,000 samples are generated for each resolution, and metrics are computed using the ADM evaluation toolkit Dhariwal & Nichol (2021).

**Text-to-Image Generation.** We further evaluate TCA-DiT on PixArt-$\alpha$-512 Chen et al. (2024) for text-to-image synthesis. Images are generated using a 20-step DPM solver Lu et al. (2022) and CFG scale 4.5. Following ViDiT-Q Zhao et al. (2025), we report *FID* for fidelity, *ClipScore* Hessel et al. (2021) for text-image alignment, and *ImageReward* Xu et al. (2023) as a proxy for human preference. Metrics are computed on the first 1,024 COCO captions.

**Baselines.** Detailed re-implementation settings for each baseline are provided in Sec. C.2.

**Quantization Configurations.** We consider two bit-width settings: W4A8 and the more challenging W4A4. Both weights and activations are quantized using uniform asymmetric group quantization, with weights quantized offline and activations quantized dynamically at inference. Following prior works (Chen et al., 2025; Zhao et al., 2025), conditioning MLPs are kept in full precision due to negligible cost. The two matrix multiplications in self-attention are quantized to the same bit-width as activations. Complete hyperparameters and additional details are provided in Sec. C.1.

**Hardware Implementation Details.** For deployment, we adopt the open-source GEMM CUDA kernels from ViDiT-Q. Consistent with its design, *kernel fusion* Wang et al. (2010) integrates dynamic quantization operators and the learned rotation matrix $R$ into preceding operations such as LayerNorm, GeLU, and residual connections. Scaling factors $s$ are fused into earlier layers following SmoothQuant Xiao et al. (2023). Channel reordering is incorporated into LayerNorm following RPTQ Yuan et al. (2023), with weight outputs reordered offline to match the optimal permutation

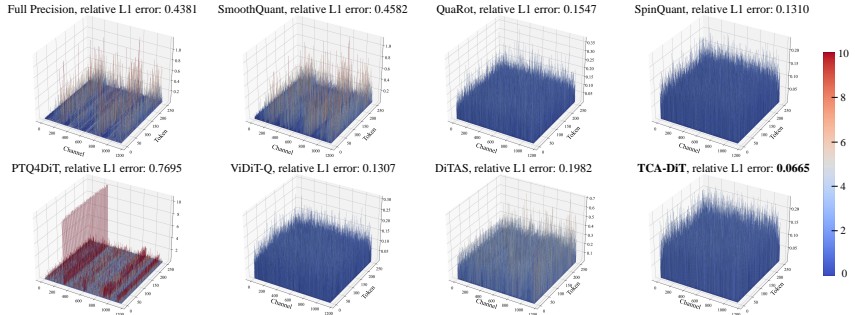

Figure 7: Activation distribution comparison between FP, baselines, and TCA-DiT.

of the subsequent linear layer. These fusions eliminate redundant operations, introducing negligible overhead. Efficiency is evaluated in terms of model size, memory usage, and latency. All experiments are conducted on an NVIDIA A6000 GPU (48GB) with CUDA 12.4.

## 4.2 QUANTIZATION PERFORMANCE

**Class-conditioned Image Generation.** Fig. 6 reports results on DiT-XL/2 with ImageNet 256×256. Under W4A8, TCA-DiT attains near-FP performance and consistently surpasses all baselines. With 100 and 50 denoising steps, it improves FID by **0.54** and **1.66** over the strongest competitor, while also yielding higher IS and Precision. Under the more challenging W4A4 setting, where most methods fail, TCA-DiT remains stable. For example, at 50 steps, Q-DiT exhibits an FID increase of 28.16 over FP, while PTQ4DiT and DiTAS collapse into blank or noisy outputs (thus omitted). In contrast, TCA-DiT achieves the best results, highlighting its robustness to timestep-dependent anomalies. Qualitative results further confirm this: under W4A4 with 100 steps, TCA-DiT preserves fine structures and realistic details, whereas competing methods produce distorted or implausible images. Among rotation-based methods, SpinQuant generally outperforms QuaRot by

Table 1: Performance comparison of text-to-image generation on COCO.

| Bit-width (W/A) | Method | FID ↓ | CLIP ↑ | IR ↑ |
|---|---|---|---|---|
| 16/16 | FP | - | 0.254 | 0.801 |
| 4/8 | SmoothQuant | 42.54 | 0.234 | 0.643 |
| | QuaRot | 37.75 | 0.246 | 0.711 |
| | SpinQuant | 36.84 | 0.247 | 0.718 |
| | PTQ4DiT | 75.10 | 0.218 | 0.517 |
| | Q-DiT | 42.85 | 0.233 | 0.649 |
| | ViDiT-Q | 40.38 | 0.248 | 0.663 |
| | DiTAS | 58.98 | 0.204 | 0.569 |
| | **Ours** | **33.82** | **0.254** | **0.772** |
| 4/4 | SmoothQuant | 241.50 | 0.163 | -2.262 |
| | QuaRot | 45.35 | 0.224 | 0.585 |
| | SpinQuant | 45.12 | 0.226 | 0.601 |
| | PTQ4DiT | 370.24 | 0.132 | -2.272 |
| | Q-DiT | 140.46 | 0.204 | -1.760 |
| | ViDiT-Q | 70.84 | 0.245 | 0.404 |
| | DiTAS | 127.45 | 0.194 | -1.256 |
| | **Ours** | **41.38** | **0.253** | **0.769** |

learning task-driven rotations, yet rotation or scaling (SmoothQuant) alone fails to handle timestep-varying anomalies. By jointly integrating ARC, ATG, and RGQ, TCA-DiT effectively suppresses quantization distortion and sustains generation quality across bit-widths and timesteps. Fig. 7 further visualizes activation distributions at `blocks.1.mlp.fc1` under W4A4 with 100 steps. TCA-DiT produces more uniform distributions and achieves the lowest quantization error, consistent with its fidelity gains from explicit anomaly alignment. Additional results at 512×512 and qualitative examples in Secs. D and G.2 further confirm its scalability and effectiveness.

**Text-to-image Generation.** Tab. 1 reports results on PixArt-$\alpha$-512. Under W4A8, TCA-DiT improves FID by **3.02** over the strongest baseline SpinQuant and by **6.56** over the previous SOTA ViDiT-Q, while also achieving the best CLIP and ImageReward scores. In the more challenging W4A4 setting, where other methods suffer severe degradation or even negative ImageReward, TCA-DiT still reduces FID by **29.46** over ViDiT-Q and consistently yields higher alignment and preference scores. Moreover, it better preserves fine-grained prompt-image correspondence, whereas rotation-only and scaling-only baselines often drift from the intended content. Overall, TCA-DiT delivers robust quality and faithful alignment under extreme quantization, with qualitative results in Sec. G.2 further highlighting its detail preservation and semantic consistency.

**Memory saving and speedup.** Fig. 9 reports model size, runtime memory, and inference latency of PixArt-$\alpha$ (batch size 1, 20 denoising steps). As shown in Fig. 9(a), TCA-DiT compresses the FP16 model from 2.45GB to 0.63GB under both W4A8 and W4A4—a 3.8× reduction despite retaining several FP modules (e.g., conditioning MLPs). In Fig. 9(b), runtime memory drops from 1.73GB

| Method | FID ↓ | sFID ↓ |
|---|---|---|
| FP | 5.64 | 18.95 |
| DQ-G128 | 28.79 | 89.15 |
| QuaRot | 9.96 | 24.61 |
| QuaRot-SQuni | 9.93 | 24.64 |
| **L-Rec** | 9.82 | 24.52 |
| **L-ARC+SQuni** | 9.32 | 24.27 |
| **L-ARC+SQATG** | 9.19 | 24.15 |
| **Full TCA-DiT** | **9.04** | **24.01** |

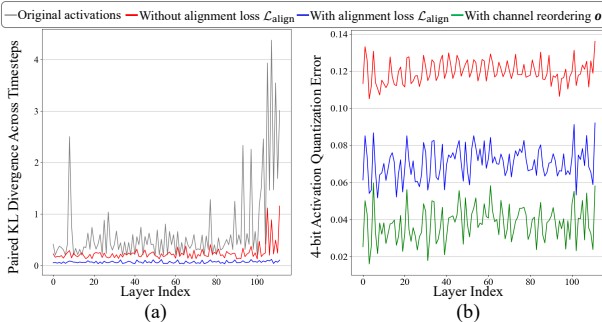

Figure 8: Ablation study on ImageNet 256×256 with W4A4 and 100 denoising steps. **Left:** Quantitative results of different ablated variants. **Right:** Layer-wise analyses. (a) Incorporating the anomaly alignment loss $\mathcal{L}_{align}$ sharply reduces the KL divergence of activation distributions across timesteps, indicating more consistent channel alignment. (b) Channel reordering further decreases quantization error, complementing anomaly alignment for robust low-bit quantization.

in FP16 to 2.6× and 3.8× smaller footprints under W4A8 and W4A4, respectively. Latency in Fig. 9(c) further highlights efficiency: starting at 31ms for FP16, TCA-DiT achieves a 1.7× speedup at W4A8. Compared to a naive W4A8 baseline without dynamic quantization, group quantization, and anomaly suppression, TCA-DiT adds negligible overhead (1.8× vs. 1.7×). With optimized W4A4 kernels, speedup reaches 3.5×, confirming the practicality and scalability of our design.

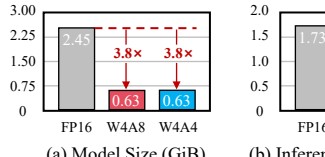
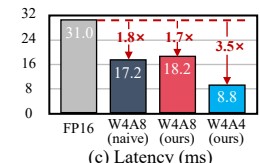

Figure 9: Efficiency analysis of TCA-DiT in model size, memory usage, and inference latency.

## 4.3 ABLATION STUDIES

We evaluate the contribution of each TCA-DiT component on DiT-XL/2 with ImageNet 256×256 under W4A4 setting with 100 denoising steps, as shown in Fig. 8. Starting from a baseline with dynamic activation quantization and group quantization (**DQ-G128**, group size $g = 128$), FID is 28.79. Adding random rotations (**QuaRot**) effectively suppresses outliers, reducing FID to 9.96 (↓18.83). Combining it with SmoothQuant on 25 uniformly sampled timesteps (**QuaRot+SQuni**) yields little improvement (9.93 vs. 9.96), indicating that random rotations cannot align anomaly channels across timesteps, thus limiting scaling accuracy. Learning rotations with $\mathcal{L}_{rec}$ (**L-Rec**) further lowers FID to 9.82 (↓0.11), suggesting better adaptation to activation distributions. Adding $\mathcal{L}_{align}$ (**L-ARC+SQuni**) gives a large gain, lowering FID to 9.32 (↓0.50). As shown in Fig. 4, channels become consistently aligned across timesteps, stabilizing scaling factors and suppressing residual anomalies. Quantitatively, Fig. 8-(a) shows a sharp drop in KL divergence between timesteps, approaching zero in most layers. Replacing uniform sampling with clustering-based calibration (**L-ARC+SQATG**) further improves FID to 9.19 (↓0.13), as selected timesteps better capture distributional diversity along the denoising trajectory. Finally, introducing RGQ (**Full TCA-DiT**) achieves the best FID of 9.04 (↓0.15). As shown in Fig. 8-(b), while $\mathcal{L}_{align}$ already reduces quantization error, RGQ provides complementary gains by mitigating intra-group variance. Overall, these results confirm that anomaly alignment and channel reordering act synergistically, jointly enabling robust low-bit quantization.

## 5 CONCLUSION

We present **TCA-DiT**, a novel PTQ framework for Diffusion Transformers. By jointly integrating Anomaly-aware Rotation Calibration (ARC), Anomaly-guided Timestep Grouping (ATG), and Reordered Group Quantization (RGQ), TCA-DiT aligns anomaly channels across timesteps and alleviates channel heterogeneity, enabling stable and robust low-bit quantization. Experiments on DiT-XL/2 and PixArt-$\alpha$ show that TCA-DiT not only surpasses prior PTQ baselines in fidelity and text-image alignment, but also achieves up to 3.8× memory savings and 3.5× speedup. These results establish anomaly alignment as a key principle for efficient, high-quality DiT quantization.

REPRODUCIBILITY STATEMENT

We have taken extensive measures to facilitate the reproducibility of our work. An anonymized implementation code is included in the supplementary materials, together with the generated samples. The main text and appendix provide detailed descriptions of the algorithms, optimization procedures, ablation studies, all hyperparameter settings, and experimental protocols to ensure transparency. With these resources, independent researchers should be able to replicate our results and validate the claims presented in this paper.

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

## A  DERIVATION OF LAYER RECONSTRUCTION LOSS

In this work, we repeatedly employ the **layer reconstruction loss** as a surrogate objective to approximate the task loss. For example, in ARC (Sec. 3.1), the rotation matrix $\boldsymbol{R}$ of each layer is optimized by minimizing Eq. (2), while in RGQ (Sec. 3.3), Eq. (8) serves as the fitness function in evolutionary search to identify optimal channel permutations. The layer reconstruction loss is defined as:

$$\min_{Q_{\boldsymbol{W}^{(l)}}, Q_{\boldsymbol{X}^{(l)}}} \mathcal{L}_{\text{rec}}(\boldsymbol{W}^{(l)}, \boldsymbol{X}^{(l)}) = \|\boldsymbol{X}_q^{(l)}\boldsymbol{W}_q^{(l)} - \boldsymbol{X}^{(l)}\boldsymbol{W}^{(l)}\|_2^2, \tag{9}$$

where $\boldsymbol{W}^{(l)}$ and $\boldsymbol{X}^{(l)}$ denote the weights and inputs of the $l$-th layer, and $\boldsymbol{W}_q^{(l)}$, $\boldsymbol{X}_q^{(l)}$ are their quantized counterparts. $Q(\cdot)$ is the uniform quantization function defined in Eq. (1). As shown in Sec. 1 (see Fig. 5b), layer reconstruction loss exhibits strong empirical correlation with task performance, which justifies its widespread use as a surrogate objective in our framework. To further support this observation, we provide a brief derivation illustrating why minimizing per-layer reconstruction loss serves as a reasonable approximation to minimizing the task loss. This derivation follows the analysis in AdaRound Nagel et al. (2020), with a more detailed discussion available therein.

Assuming weight-only quantization. Let $\mathbf{w}$ be the flattened weight vector of the network. Quantization introduces a perturbation defined as:

$$\Delta\mathbf{w} = Q_{\mathbf{w}}(\mathbf{w}) - \mathbf{w} = \mathbf{w}_q - \mathbf{w}, \tag{10}$$

where $\mathbf{w}_q$ is the quantized version of $\mathbf{w}$. Let $\mathcal{L}(\mathbf{x}, \mathbf{y}, \mathbf{w})$ denote the task loss. The quantization objective can then be expressed as:

$$\arg\min_{\Delta\mathbf{w}} \mathbb{E}[\mathcal{L}(\mathbf{x}, \mathbf{y}, \mathbf{w} + \Delta\mathbf{w}) - \mathcal{L}(\mathbf{x}, \mathbf{y}, \mathbf{w})]. \tag{11}$$

Directly evaluating Eq. (11) requires a full forward pass for each perturbation $\Delta\mathbf{w}$, which is computationally infeasible. To address this, we approximate the loss difference using a *second-order*

*Taylor expansion*:

$$\mathbb{E}\left[\mathcal{L}(\mathbf{x}, \mathbf{y}, \mathbf{w} + \Delta\mathbf{w}) - \mathcal{L}(\mathbf{x}, \mathbf{y}, \mathbf{w})\right] \approx \mathbb{E}\left[\Delta\mathbf{w}^\top \nabla_\mathbf{w}\mathcal{L}(\mathbf{x}, \mathbf{y}, \mathbf{w}) + \frac{1}{2}\Delta\mathbf{w}^\top \nabla_\mathbf{w}^2\mathcal{L}(\mathbf{x}, \mathbf{y}, \mathbf{w})\Delta\mathbf{w}\right] \tag{12}$$

$$= \Delta\mathbf{w}^\top \mathbf{g}^{(\mathbf{w})} + \frac{1}{2}\Delta\mathbf{w}^\top \boldsymbol{H}^{(\mathbf{w})}\Delta\mathbf{w} \tag{13}$$

where $\mathbf{g}^{(\mathbf{w})}$ and $\boldsymbol{H}^{(\mathbf{w})}$ are the expected gradient and Hessian of the task loss:

$$\mathbf{g}^{(\mathbf{w})} = \mathbb{E}\left[\nabla_\mathbf{w}\mathcal{L}(\mathbf{x}, \mathbf{y}, \mathbf{w})\right], \tag{14}$$

$$\boldsymbol{H}^{(\mathbf{w})} = \mathbb{E}\left[\nabla_\mathbf{w}^2\mathcal{L}(\mathbf{x}, \mathbf{y}, \mathbf{w})\right]. \tag{15}$$

For tractability, we assume negligible cross-layer dependencies, i.e., $\boldsymbol{H}^{(\mathbf{w})}$ is block-diagonal with each block corresponding to one layer. This yields the per-layer optimization problem:

$$\underset{\Delta\boldsymbol{W}^{(l)}}{\arg\min} \; \mathbb{E}\left[\mathbf{g}^{(\boldsymbol{W}^{(l)})^\top}\Delta\boldsymbol{W}^{(l)} + \frac{1}{2}\Delta\boldsymbol{W}^{(l)^\top}\boldsymbol{H}^{(\boldsymbol{W}^{(l)})}\Delta\boldsymbol{W}^{(l)}\right]. \tag{16}$$

For a converged pretrained model, the first-order term is negligible, reducing the objective to:

$$\underset{\Delta\boldsymbol{W}^{(l)}}{\arg\min} \; \mathbb{E}\left[\Delta\boldsymbol{W}^{(l)^\top}\boldsymbol{H}^{(\boldsymbol{W}^{(l)})}\Delta\boldsymbol{W}^{(l)}\right]. \tag{17}$$

**Structure of the Hessian.** Consider two weights within the same layer:

$$\frac{\partial^2\mathcal{L}}{\partial\boldsymbol{W}_{i,j}^{(l)}\partial\boldsymbol{W}_{m,o}^{(l)}} = \frac{\partial}{\partial\boldsymbol{W}_{m,o}^{(l)}}\left[\frac{\partial\mathcal{L}}{\partial\mathbf{z}_i^{(l)}}\mathbf{x}_j^{(l)}\right] = \frac{\partial^2\mathcal{L}}{\partial\mathbf{z}_i^{(l)}\partial\mathbf{z}_m^{(l)}}\mathbf{x}_j^{(l)}\mathbf{x}_o^{(l)}, \tag{18}$$

where $\mathbf{z}^{(l)} = \boldsymbol{W}^{(l)}\mathbf{x}^{(l)}$ are the pre-activations of layer $l$ and $\mathbf{x}^{(l)}$ is its input. In matrix form:

$$\boldsymbol{H}^{(\boldsymbol{W}^{(l)})} = \mathbb{E}\left[\mathbf{x}^{(l)}\mathbf{x}^{(l)^\top} \otimes \nabla_{\mathbf{z}^{(l)}}^2\mathcal{L}\right], \tag{19}$$

where $\otimes$ denotes the Kronecker product, and $\nabla_{\mathbf{z}^{(l)}}^2\mathcal{L}$ is the Hessian of the task loss w.r.t. the pre-activations.

**Simplifying assumptions.** To avoid expensive second-order backpropagation, we further assume $\nabla_{\mathbf{z}^{(l)}}^2\mathcal{L}$ is diagonal:

$$\boldsymbol{H}^{(\boldsymbol{W}^{(l)})} = \mathbb{E}\left[\mathbf{x}^{(l)}\mathbf{x}^{(l)^\top} \otimes \mathrm{diag}(\nabla_{\mathbf{z}^{(l)}}^2\mathcal{L}_{i,i})\right]. \tag{20}$$

Substituting Eq. (20) into Eq. (17), the problem decomposes across output channels:

$$\underset{\Delta\boldsymbol{W}_{k,:}^{(l)}}{\arg\min} \; \mathbb{E}\left[\nabla_{\mathbf{z}^{(l)}}^2\mathcal{L}_{k,k} \cdot \Delta\boldsymbol{W}_{k,:}^{(l)}\mathbf{x}^{(l)}\mathbf{x}^{(l)^\top}\boldsymbol{W}_{k,:}^{(l)^\top}\right] \tag{21}$$

$$\overset{(a)}{=} \underset{\Delta\boldsymbol{W}_{k,:}^{(l)}}{\arg\min} \; \Delta\boldsymbol{W}_{k,:}^{(l)}\mathbb{E}\left[\mathbf{x}^{(l)}\mathbf{x}^{(l)^\top}\right]\Delta\boldsymbol{W}_{k,:}^{(l)^\top} \tag{22}$$

$$= \underset{\Delta\boldsymbol{W}_{k,:}^{(l)}}{\arg\min} \; \mathbb{E}\left[\left(\Delta\boldsymbol{W}_{k,:}^{(l)}\mathbf{x}^{(l)}\right)^2\right], \tag{23}$$

where step (a) assumes $\nabla_{\mathbf{z}^{(l)}}^2\mathcal{L}_{i,i} = c_i$ is approximately constant across the data distribution.

**Connection to Layer Reconstruction Loss.** The final form in Eq. (23) matches the layer reconstruction objective in Eq. (9), except that it does not explicitly include activation quantization. To account for accumulated quantization errors and to jointly optimize weights and activations, we adopt Eq. (9) as the ultimate proxy for the task loss.

---

**Algorithm 2** Evolutionary Search for Optimal Channel Permutation

---

**Require:** Linear layer $f(\cdot, \hat{W})$ with transformed weight $\hat{W}$ and original weight $W$; calibration data $\{X^t\}_{t=1}^T$
    and their transformed counterparts $\{\hat{X}^t\}_{t=1}^T$; population size $S$; number of iterations $N$; mutation proba-
    bility $p$; number of elites $k$; tournament size $w$.
1: Initialize population $\mathcal{P} = \{o^j\}_{j=1}^S$, where each $o^j$ is a random permutation of $\{1, \ldots, \mathcal{C}_i\}$.
2: **for** $t = 1$ to $N$ **do**
3:     Evaluate fitness of each $o^j \in \mathcal{P}$ using layer reconstruction loss $\mathcal{L}_{\text{rec}}$ in Eq. (8).
4:     Create new population $\mathcal{P}^{\text{new}} \leftarrow \emptyset$.
5:     Add top-$k$ elites (lowest $\mathcal{L}_{\text{rec}}$) from $\mathcal{P}$ to $\mathcal{P}^{\text{new}}$.
6:     **while** $|\mathcal{P}^{\text{new}}| < S$ **do**
7:         Select parent $o^m$ via Tournament$(\mathcal{P}, w)$.
8:         Select parent $o^n$ via Tournament$(\mathcal{P}, w)$.
9:         $o_{\text{cross}} \leftarrow$ CrossOver$(o^m, o^n)$.
10:        $o_{\text{mutate}} \leftarrow$ Mutate$(o_{\text{cross}})$ with probability $p$.
11:        Append $o_{\text{mutate}}$ to $\mathcal{P}^{\text{new}}$.
12:     $\mathcal{P} \leftarrow \mathcal{P}^{\text{new}}$.
13: **return** Best permutation $o^\star$ (lowest $\mathcal{L}_{\text{rec}}$); apply $o^\star$ to reorder activations and weights before group quan-
    tization.

---

# B  TCA-DiT Pipeline

## B.1  Evolutionary Search Algorithm

To identify an optimal channel permutation for group quantization, we adopt an **evolutionary search strategy** (Alg. 2). The algorithm maintains a population of candidate permutations, which are iteratively refined through selection, crossover, and mutation, with the **layer reconstruction loss** (Eq. (8)) serving as the fitness objective.

Two implementation refinements are particularly important. First, for crossover, we adopt the *Order Crossover (OX)* operator, ensuring that offspring remain a valid permutation after recombination, thereby preserving the integrity of channel orderings. Second, during mutation, we restrict random swaps to channel indices whose absolute difference exceeds the group size $g$ (set to 128 in our experiments). This prevents ineffective swaps within the same group, which would otherwise yield no change under subsequent group quantization. Together, these tailored operations improve search efficiency and enable more reliable convergence toward permutations that minimize reconstruction error.

## B.2  Overall Procedure of TCA-DiT

We now present the complete pipeline of **TCA-DiT**, which integrates the three core components described in Secs. 3.1 to 3.3. As illustrated in Alg. 3, each linear layer of the pre-trained DiT is calibrated and quantized through the following five stages:

**(1) Anomaly-guided Timestep Grouping (ATG):** Instead of calibrating on all timesteps, we cluster timesteps by their activation distributions and select representative steps to form a compact calibration set. This substantially reduces cost while preserving distributional fidelity.

**(2) Anomaly-aware Rotation Calibration (ARC):** We apply an orthogonal rotation $R$ followed by channel scaling $s$ to align outlier channels and suppress residual anomalies across timesteps. The rotation is optimized via a joint reconstruction–alignment loss, ensuring that temporally unstable channels are aligned before scaling.

**(3) Reordered Group Quantization (RGQ):** To mitigate remaining channel misalignment, we perform an evolutionary search to find an optimal channel permutation prior to group quantization. This reduces intra-group variance and lowers quantization error.

**(4) Offline Fusion and Weight Quantization:** The learned transforms ($R$, $s$, and channel permutation) are fused into adjacent operators such as LayerNorm, eliminating runtime overhead. Groupwise weight quantization is then applied using precomputed quantization parameters.

---

**Algorithm 3 TCA-DiT**: **T**emporal **C**hannel **A**lignment for **Di**ffusion **T**ransformers

---

**Require:** Pre-trained DiT $\mathcal{F}(\Theta) = f^L_{\boldsymbol{W}^L} \circ \cdots \circ f^1_{\boldsymbol{W}^1}$; total timesteps $T$; number of timestep groups $G \ll T$; group size $g$; bit-widths $(b_{\boldsymbol{W}}, b_{\boldsymbol{X}})$; ARC hyperparameters $(\alpha, \beta, \gamma)$; evolutionary search hyperparameters $(S, N, p, k, w)$.

**Ensure:** Quantized DiT $\mathcal{F}_q(\Theta)$ with fused transforms and negligible runtime overhead.

1: **for** $l = 1$ to $L$ **do**               ▷ Process each linear layer $f^l_{\boldsymbol{W}^l}$

2:   Extract weight $\boldsymbol{W} = \boldsymbol{W}^l$ and collect activations $\{\boldsymbol{X}^t\}_{t=1}^T$ of the current layer through one forward pass of $\mathcal{F}(\Theta)$.

3:   **(A) Anomaly-guided Timestep Grouping (ATG)**

4:   Compute normalized per-channel distributions $\{\boldsymbol{q}^t\}_{t=1}^T$.

5:   Apply constrained hierarchical clustering (Alg. 1) to partition timesteps into $G$ groups with boundaries $\{\tau_i\}_{i=1}^{G-1}$ and representatives $\{t_g\}_{g=1}^G$.

6:   Construct the compact calibration set $\{\boldsymbol{X}^{t_g}\}_{g=1}^G$.

7:   **(B) Anomaly-aware Rotation Calibration (ARC)**

8:   Initialize rotation $\boldsymbol{R}^l$ (e.g., randomized Hadamard).

9:   Optimize $\boldsymbol{R}^l$ via *Cayley-SGD* with objective:

$$\min_{\boldsymbol{R}^l} \underbrace{\frac{1}{G}\sum_{g=1}^G \|Q(\tilde{\boldsymbol{X}}^{t_g})Q^\top(\tilde{\boldsymbol{W}}) - \boldsymbol{X}^{t_g}\boldsymbol{W}^\top\|_2^2}_{\mathcal{L}_{\text{rec}}} + \beta \underbrace{\frac{1}{G}\sum_{g \neq g_{\text{mid}}} D_{\text{KL}}(\boldsymbol{p}^{t_g}\|\boldsymbol{p}^{t_{g_{\text{mid}}}})}_{\mathcal{L}_{\text{align}}},$$

where $\tilde{\boldsymbol{X}}^t = \boldsymbol{X}^t\boldsymbol{R}^l$, $\tilde{\boldsymbol{W}} = \boldsymbol{W}\boldsymbol{R}^l$, and $t_{g_{\text{mid}}}$ denotes the middle-group representative.

10:   Estimate scaling factor $\boldsymbol{s}^l$ via Eq. (5); obtain $\hat{\boldsymbol{X}}^{t_g} = \tilde{\boldsymbol{X}}^{t_g}\text{diag}(\boldsymbol{s}^l)^{-1}$, $\hat{\boldsymbol{W}} = \tilde{\boldsymbol{W}}\text{diag}(\boldsymbol{s}^l)$.

11:   **(C) Reordered Group Quantization (RGQ)**

12:   Run evolutionary search (Alg. 2) on $\{\hat{\boldsymbol{X}}^{t_g}\}$ to identify the optimal permutation $\boldsymbol{o}^\star$ minimizing reconstruction loss (Eq. (8)).

13:   Define $\bar{\boldsymbol{W}} = \hat{\boldsymbol{W}}_{:, \boldsymbol{o}^\star}$, $\bar{\boldsymbol{X}}^{t_g} = \hat{\boldsymbol{X}}^{t_g}_{:, \boldsymbol{o}^\star}$.

14:   **(D) Offline Fusion & Weight Quantization**

15:   Fuse $\boldsymbol{R}^l$, $\text{diag}(\boldsymbol{s}^l)$, and $\boldsymbol{o}^\star$ into adjacent operators (e.g., LayerNorm) to eliminate runtime overhead.

16:   Apply group-wise weight quantization: $\boldsymbol{W}^{q,l} = Q_{\boldsymbol{W}}(\bar{\boldsymbol{W}})$ with precomputed per-group scales and zero-points.

17:   **(E) Dynamic Activation quantization**

18:   During inference, compute activation quantization parameters (scale/zero-point) dynamically from each mini-batch.

19:   Fuse dynamic quantization operators with adjacent layers (linear/normalization) via kernel fusion.

20: **return** $\mathcal{F}_q(\Theta)$ with statically quantized weights and dynamically quantized activations, supported by fused transforms.

---

**(5) Dynamic Activation Quantization:** Activations are quantized per mini-batch with dynamic ranges, and the corresponding operators are fused into neighboring kernels, ensuring efficient inference with negligible overhead.

Together, these steps yield a quantized DiT with weights that are statically quantized and activations that are dynamically quantized, with efficiency maintained through systematic kernel fusion.

## C  IMPLEMENTATION DETAILS

### C.1  IMPLEMENTATION DETAILS OF TCA-DIT

**DiT-XL/2.** For class-conditioned image generation with DiT-XL/2 Peebles & Xie (2023), we construct the calibration set by randomly sampling 32 class labels from 1,000 ImageNet classes. For each label, we generate one image using a 100-step DDPM solver Ho et al. (2020) and record outputs at all timesteps, yielding 3,200 calibration samples in total. For Anomaly-guided Timestep Grouping (ATG), we set the number of timestep groups to $G = 10$ under both $T = 100$ and $T = 50$ denoising schedules. One representative timestep is chosen per group, reducing the calibration set to 320 samples. For Anomaly-aware Rotation Calibration (ARC), we set the scaling factor $\gamma = 20$ (Eq. (3)). We observe that the method is robust to $\gamma$, with values in $[5, 50]$ producing comparable results. The trade-off coefficient $\beta$ (Eq. (4)) is fixed at 0.05 to balance reconstruction and alignment

losses. Following SpinQuant Liu et al. (2025), rotation matrices $\boldsymbol{R}$ are optimized via *Cayley SGD* Li et al. (2020) for 400 iterations with a linearly decayed learning rate from 1.5 to 0. The scaling factor $\alpha$ (Eq. (5)) is set to 0.5, which we find sufficient without additional grid search. For Reordered Group Quantization (RGQ), we use a group size of $g = 128$. The evolutionary search (Alg. 2) is configured with population size $\mathcal{S} = 50$, iterations $N = 20$, mutation probability $p = 0.8$, elite size $k = 3$, and tournament size $w = 5$. The same configuration is applied to both ImageNet $256 \times 256$ and $512 \times 512$. All quantization and evaluation experiments are performed on a single NVIDIA A6000 GPU (48GB).

**PixArt-$\alpha$-512.** For text-to-image generation with PixArt-$\alpha$-512 Chen et al. (2024), we construct the calibration set by randomly sampling 32 text prompts from COCO captions. For each prompt, we generate a single image with the full-precision model using a 20-step DPM solver Lu et al. (2022), collecting outputs at all timesteps to obtain 640 calibration samples. Since the denoising schedule consists of only 20 steps, this set already captures sufficient temporal variations, and we therefore omit ATG. All other hyperparameters, including those for ARC and RGQ, follow the DiT-XL/2 configuration.

A detailed hyperparameter sensitivity analysis is provided in Sec. D.3.

## C.2 Baseline Methods Implementations

In our experiments, we compare **TCA-DiT** against seven representative Post-Training Quantization methods. These include three approaches originally developed for transformer models in Natural Language Processing—**SmoothQuant** Xiao et al. (2023), **QuaRot** Ashkboos et al. (2024), and **SpinQuant** Liu et al. (2025)—which we adapt to DiT architectures based on their official implementations, as well as four recent methods specifically developed for DiTs—**PTQ4DiT** Wu et al. (2024), **Q-DiT** Chen et al. (2025), **ViDiT-Q** Zhao et al. (2025), and **DiTAS** Dong & Zhang (2025)—which we produce strictly according to their published implementations.

- **SmoothQuant.** We uniformly sample 25 timesteps (for PixArt-$\alpha$, we use the whole 20 denoising steps) across the denoising trajectory, generating 16 samples at each step to form the calibration set. The scaling factor $\boldsymbol{s}$ is computed by taking the maximum across both the timestep and sample dimensions. Layer-wise $\alpha$ is set to 0.5 by default without additional grid search.

- **QuaRot.** Following the original design, we apply a random Hadamard rotation matrix to each linear layer to balance the activations.

- **SpinQuant.** Consistent with the official setup, 800 samples are collected from the dataset. Rotation matrices $\boldsymbol{R}$ are optimized using the original DiT denoising loss and trained via *Cayley SGD* Li et al. (2020) with a learning rate linearly decayed from 1.5 to 0 over 100 iterations.

- **PTQ4DiT.** We adopt Channel-wise Salience Balancing (CSB) and Spearman's $\rho$-guided Salience Calibration (SSC) to balance activations. As in the original work, *block reconstruction* Li et al. (2021) is used to optimize quantization parameters jointly with the AdaRound Nagel et al. (2020) $\alpha$ parameter. Static per-tensor quantization is applied to both weights and activations.

- **Q-DiT.** We follow the original design by applying group quantization to both weights and activations, where activations are dynamically quantized. Group sizes are selected per layer using the proposed evolutionary search.

- **ViDiT-Q.** We adopt its Static-Dynamic Channel Balancing strategy for activation balancing. For both W4A8 and W4A4 settings, we use the proposed mixed-precision quantization strategy with Mean Squared Error (MSE) as the sensitivity indicator. Weights are quantized per output channel, while activations are quantized dynamically per token.

- **DiTAS.** We implement its Temporal-aggregated Smoothing to mitigate the time-varying activations imbalance, combined with Grid Search Optimization to determine optimal $\alpha$ values. Following the original design, weights are quantized per input channel, and activations are quantized dynamically per token.

To ensure a fair comparison and isolate the benefits of integrating rotation- and scale-based techniques, we adapt SmoothQuant, QuaRot, and SpinQuant to employ the same group-wise quantization scheme as TCA-DiT, using a group size of $g = 128$ for both weights and activations, and dynamic quantization for activations.

# D  ADDITIONAL EXPERIMENTAL RESULTS

## D.1  QUANTIZATION TIME COMPARISON

Table 2: Quantization time comparison across different models under W4A8.

| Method | DiT-XL/2-256 | DiT-XL/2-512 | PixArt-$\alpha$-512 |
|---|---|---|---|
| SmoothQuant | 0h 2m | 0h 5m | 0h 7m |
| QuaRot | 0h 1m | 0h 1m | 0h 2m |
| SpinQuant | 0h 24m | 1h 24m | 2h 31m |
| PTQ4DiT | 6h 27m | 28h 32m | 32h 30m |
| Q-DiT | 102h 37m | 333h 11m | 566h 24m |
| ViDiT-Q | 0h 8m | 0h 29m | 0h 54m |
| DiTAS | 1h 32m | 5h 22m | 9h 40m |
| **TCA-DiT (Ours)** | 2h 12m | 7h 43m | 11h 32m |

We compare the end-to-end quantization time of TCA-DiT against all baselines across DiT-XL/2 and PixArt-$\alpha$-512, as summarized in Tab. 2. 'DiT-XL/2-256' and 'DiT-XL/2-512' refer to quantization on ImageNet 256×256 and 512×512, respectively. For fairness, the time required to construct the calibration set is excluded. All experiments are conducted under the W4A8 setting on a single NVIDIA A6000 GPU (48 GB).

Taking DiT-XL/2-256 as an example, we analyze the runtime characteristics of each method. **SmoothQuant** relies only a single forward pass over the calibration set to collect activation statistics and compute scaling factors, resulting in a fast runtime of 2 minutes. **QuaRot**, which applies random Hadamard rotations to weights and activations independent of calibration data, is even faster (1 minute). **SpinQuant** optimizes rotation matrices using a denoising loss, leading to moderate cost (24 minutes). **PTQ4DiT** combines lightweight salience balancing with a highly expensive block reconstruction procedure, in which per-block rounding parameters are iteratively optimized, yielding a runtime of 6 hours 27 minutes. **Q-DiT** is dominated by its evolutionary search that allocates layer-wise group sizes based on FID. Each candidate solution requires sampling hundreds of images and computing distributional distances, resulting in a prohibitive runtime exceeding 100 hours. **ViDiT-Q** integrates QuaRot and SmoothQuant with mixed-precision bit allocation guided by MSE sensitivity. While requiring extra calibration passes, it remains efficient (8 minutes). **DiTAS** introduces temporal smoothing at a cost similar to SmoothQuant, but also conducts a grid search for per-layer scaling, leading to substantially higher runtime (1 hour 32 minutes).

Our proposed **TCA-DiT** incorporates three additional procedures. First, Anomaly-guided Timestep Grouping (ATG) performs Constrained Hierarchical Clustering (Alg. 1) over activation trajectories, which is relatively efficient (∼4 minutes). Second, Anomaly-aware Rotation Calibration (ARC) optimizes rotation matrices using a joint reconstruction—alignment losses, costing about 40 seconds per layer and summing to roughly one hour. Finally, Reordered Group Quantization (RGQ) performs layer-wise evolutionary search (Alg. 2) with a modest population size and iteration budget, adding approximately another hour. The total runtime on DiT-XL/2-256 is 2 hours 12 minutes. Although slower than ViDiT-Q, this cost is acceptable given the significant improvements in fidelity and robustness (see Fig. 6 and Tabs. 1 and 3).

For DiT-XL/2-512, the quadrupled token count compared to 256 resolution proportionally increases quantization time across all methods. On PixArt-$\alpha$-512, the additional cross-attention layer in each DiT block further prolongs runtime. Despite these overheads, TCA-DiT maintains practical efficiency: while not the fastest, it achieves a favorable balance between quantization cost and task performance. Future work may explore further reducing calibration overhead while preserving accuracy.

Table 3: Performance comparison of class-conditioned image generation on ImageNet 512×512 (cfg=1.5). '(W/A)' indicates that the precision of weights and activations is W and A bits, respectively. '-' means this method produces unreadable contents.

| Timesteps | Bit-width (W/A) | Method | FID ↓ | sFID ↓ | IS ↑ | Precision ↑ |
|---|---|---|---|---|---|---|
| 100 | 32/32 | FP | 6.55 | 20.6 | 232.74 | 0.8317 |
| | 4/8 | SmoothQuant | 11.56 | 25.15 | 169.85 | 0.7706 |
| | | QuaRot | 12.18 | 24.84 | 165.96 | 0.7657 |
| | | SpinQuant | 11.58 | 23.92 | 171.40 | 0.7741 |
| | | PTQ4DiT | 19.00 | 50.71 | 121.35 | 0.7514 |
| | | Q-DiT | 8.76 | 22.85 | 197.70 | 0.7953 |
| | | ViDiT-Q | 12.08 | 26.71 | 166.94 | 0.7782 |
| | | DiTAS | 15.18 | 46.69 | 172.20 | 0.7815 |
| | | **Ours** | **8.12** | **22.17** | **208.55** | **0.7987** |
| | 4/4 | SmoothQuant | 43.87 | 89.21 | 51.22 | 0.514 |
| | | QuaRot | 26.42 | 31.97 | 102.08 | 0.6676 |
| | | SpinQuant | 24.86 | 30.67 | 108.94 | 0.6813 |
| | | PTQ4DiT | - | - | - | - |
| | | Q-DiT | 39.70 | 99.73 | 57.76 | 0.5583 |
| | | ViDiT-Q | 44.72 | 102.43 | 52.28 | 0.5228 |
| | | DiTAS | - | - | - | - |
| | | **Ours** | **18.39** | **27.58** | **153.92** | **0.7319** |
| 50 | 32/32 | FP | 7.91 | 23.01 | 214.85 | 0.7974 |
| | 4/8 | SmoothQuant | 17.83 | 31.19 | 138.66 | 0.7293 |
| | | QuaRot | 19.10 | 30.73 | 130.67 | 0.7241 |
| | | SpinQuant | 17.25 | 28.68 | 141.27 | 0.7380 |
| | | PTQ4DiT | 19.71 | 52.27 | 118.32 | 0.7336 |
| | | Q-DiT | 13.26 | 27.93 | 162.39 | 0.7644 |
| | | ViDiT-Q | 14.22 | 28.51 | 161.08 | 0.7554 |
| | | DiTAS | 18.37 | 49.20 | 149.69 | 0.7484 |
| | | **Ours** | **11.75** | **24.61** | **165.29** | **0.7785** |
| | 4/4 | SmoothQuant | 52.23 | 96.21 | 41.88 | 0.4549 |
| | | QuaRot | 38.30 | 39.62 | 77.67 | 0.6091 |
| | | SpinQuant | 35.69 | 37.24 | 90.84 | 0.6319 |
| | | PTQ4DiT | - | - | - | - |
| | | Q-DiT | 46.35 | 106.08 | 49.70 | 0.5022 |
| | | ViDiT-Q | 55.27 | 125.74 | 40.52 | 0.4810 |
| | | DiTAS | - | - | - | - |
| | | **Ours** | **24.19** | **33.93** | **131.20** | **0.6846** |

## D.2 QUANTIZATION RESULTS ON IMAGENET 512×512

Tab. 3 reports the quantization results of DiT-XL/2 on ImageNet 512×512 across different bit-widths and denoising steps. In the W4A8, TCA-DiT attains performance close to full-precision, surpassing all baselines with FID reductions of **0.64** and **1.51** at 100 and 50 timesteps, respectively. It also consistently achieves higher IS and Precision scores. Under the more challenging W4A4 setting, where most methods suffer severe degradation, TCA-DiT remains comparatively stable. At 100 timesteps, while TCA-DiT experiences a larger drop relative to the FP model (FID increases from 6.55 to 18.39, an increase of 11.84), the generated samples still exhibit high fidelity and fine-grained details (see Figs. 19 and 20), outperforming all competing baselines. These results demonstrate that, despite some degradation at extremely low bit-widths, TCA-DiT preserves robustness and delivers high-quality outputs, consistently excelling in visual quality compared to existing approaches.

## D.3 HYPERPARAMETER SENSITIVITY ANALYSIS

In this section, we examine the sensitivity of TCA-DiT to several key hyperparameters. All experiments are conducted with DiT-XL/2 on ImageNet 256×256 under W4A8, using either 100 or 50

Table 4: The effect of varying timestep group number $G$ on quantization performance and optimization overhead.

| Timesteps | $G$ | FID $\downarrow$ | sFID $\downarrow$ | Optimization Time $\downarrow$ |
|---|---|---|---|---|
| $T = 100$ | 100 | **6.13** | **19.54** | 10h 45min |
| | 25 | 6.18 | 19.71 | 3h 8min |
| | 10 | 6.20 | 19.78 | 1h 10min |
| | 5 | 6.33 | 21.26 | **0h 34min** |
| $T = 50$ | 50 | **7.48** | **22.77** | 5h 23min |
| | 10 | 7.54 | 23.07 | 1h 10min |
| | 5 | 7.68 | 24.41 | 0h 34min |
| | 2 | 8.47 | 27.59 | **0h 12min** |

denoising steps. When analyzing a specific hyperparameter, all others are fixed to the default values described in Sec. C.1.

**Timestep group number $G$.** To reduce calibration overhead, TCA-DiT employs Anomaly-guided Timestep Grouping (ATG), which partitions the $T$ denoising steps into $G$ groups and selects one representative timestep from each group to form a compact calibration set. We evaluate the trade-off between quantization performance and optimization cost as $G$ varies. Here, optimization cost refers specifically to solving the rotation matrices $\boldsymbol{R}$ in Eq. (4). As shown in Tab. 4, setting $G = T$ (i.e., no clustering, with each timestep treated independently) yields the best performance but incurs the highest optimization cost. Reducing $G$ significantly lowers the optimization cost, but it leads to a gradual degradation in generation quality. This outcome is expected, since fewer groups provide a coarser representation of the denoising trajectory, weakening the alignment of anomaly channels across timesteps. In practice, we find $G = 10$ offers a favorable balance: it maintains performance nearly identical to $G = T$ while reducing optimization time by an order of magnitude.

Table 5: The effect of varying scaling factor $\gamma$ on quantization performance.

| $\gamma$ | FID $\downarrow$ | sFID $\downarrow$ |
|---|---|---|
| 5 | 6.23 | 19.95 |
| 10 | 6.21 | 19.82 |
| 20 | **6.20** | **19.78** |
| 35 | 6.22 | 19.84 |
| 50 | 6.23 | 19.90 |

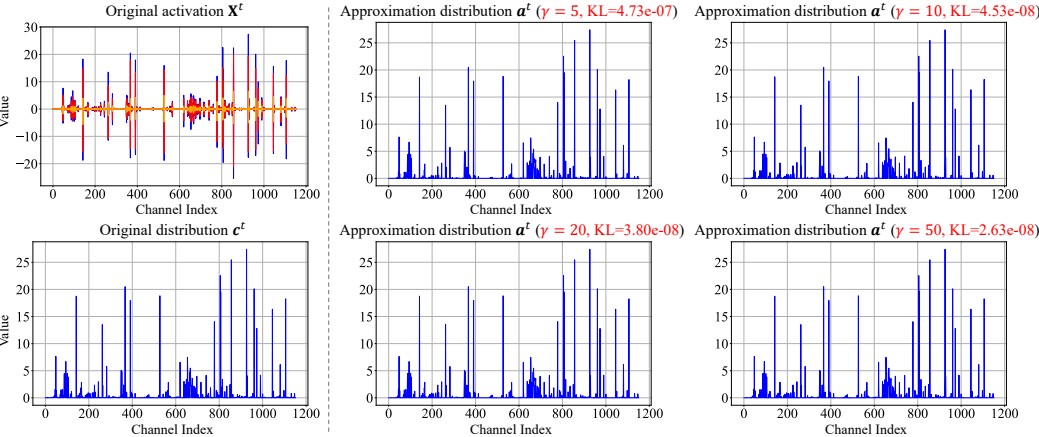

Figure 10: Original activation $\boldsymbol{X}^t$, true per-channel distribution $\boldsymbol{c}^t$, and approximated distribution $\boldsymbol{a}^t$ under different scaling factors $\gamma$.

**Scaling factor $\gamma$.** As introduced in Sec. 3.1, the **anomaly alignment loss** (Eq. (3)) is defined as:

$$\mathcal{L}_{\text{align}} = \frac{1}{T} \sum_{t=1, \ t \neq \lfloor \frac{T}{2} \rceil}^{T} D_{\text{KL}}(\boldsymbol{p}^t \| \boldsymbol{p}^{\lfloor \frac{T}{2} \rceil})$$

where $\boldsymbol{p}^t = \text{softmax}(\boldsymbol{a}^t)$. Here, $\boldsymbol{a}^t \in \mathbb{R}^{\mathcal{C}_i}$ is obtained by computing a **softmax-weighted average** of the absolute activations along each channel:

$$a_i^t = \text{SW}_\gamma\big(|\tilde{\boldsymbol{X}}_{:,i}^t|\big), \ \text{SW}_\gamma(\boldsymbol{v}) = \sum_{j=1}^{N} v_j \text{softmax}(\gamma \boldsymbol{v})_j, \tag{24}$$

where $\gamma$ is a large scaling factor. Intuitively, $\boldsymbol{a}^t$ serves as a differentiable approximation of the per-channel distribution $\boldsymbol{c}^t$, where $\boldsymbol{c}_i^t = \max(|\boldsymbol{X}_{:,i}^t|)$. A sufficiently large $\gamma$ sharpens the softmax distribution, making $\boldsymbol{a}^t$ closely approximate $\boldsymbol{c}^t$, thereby ensuring that the anomaly alignment loss is both effective and differentiable with respect to the rotation matrices $\boldsymbol{R}$. We evaluate the impact of $\gamma$ on quantization performance, with results summarized in Tab. 5. Performance remains stable across a wide range of $\gamma$ values. To further validate the approximation, we visualize $\boldsymbol{X}^t$, the ground-truth per-channel distribution $\boldsymbol{c}^t$, and its approximated $\boldsymbol{a}^t$ under varying $\gamma$ in Fig. 10, and report the corresponding KL divergence between $\boldsymbol{c}^t$ and $\boldsymbol{a}^t$. The visualizations confirm that $\boldsymbol{a}^t$ reliably tracks $\boldsymbol{c}^t$, with only minor KL divergence that decreases as $\gamma$ increases. In practice, we set $\gamma = 20$, which yields slightly improved performance while maintaining stability. These findings demonstrate that the anomaly alignment loss is robust to the choice of $\gamma$.

Table 6: The effect of varying trade-off coefficient $\beta$ on quantization performance.

| $\beta$ | FID $\downarrow$ | sFID $\downarrow$ |
|---|---|---|
| 0.001 | 6.85 | 22.69 |
| 0.01 | 6.41 | 20.74 |
| 0.05 | **6.20** | **19.78** |
| 0.1 | 6.26 | 19.96 |
| 0.5 | 6.48 | 21.24 |
| 1.0 | 6.95 | 23.51 |

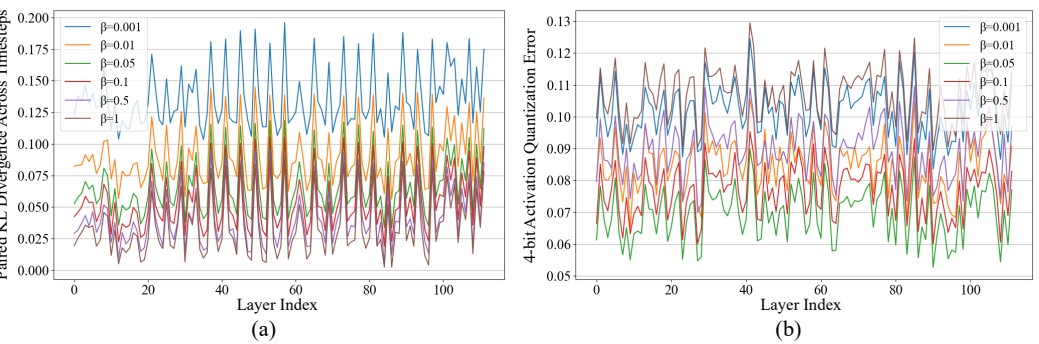

(a)              (b)

Figure 11: The effect of varying $\beta$ on per-layer (a) paired KL divergence across timesteps and (b) activation quantization error.

**Trade-off coefficient $\beta$.** As described in Sec. 3.1, the optimization objective for the rotation matrix $\boldsymbol{R}$ combines a **layer reconstruction loss** with an **anomaly alignment loss** (Eq. (4)):

$$\min_{\boldsymbol{R}} \mathcal{L} = \mathcal{L}_{\text{rec}} + \beta \cdot \mathcal{L}_{\text{align}},$$

where $\beta$ controls the balance between reconstruction fidelity and temporal anomaly alignment. We vary $\beta$ in $\{0.001, 0.01, 0.05, 0.1, 0.5, 1\}$ and report the results in Tab. 6. The best performance is observed at $\beta = 0.05$, while both smaller and larger values lead to degradation. To further under this trend, we visualize in Fig. 11 the per-layer KL divergence of rotated activations $\tilde{\boldsymbol{X}}^t$ across timesteps,

alongside the 4-bit quantization error. As shown in Fig. 11(a), increasing $\beta$ consistently reduces KL divergence, reflecting improved anomaly alignment. Intuitively, stronger alignment should facilitate more accurate scaling estimation, thereby lowering quantization error. This effect is indeed observed when $\beta \leq 0.05$, where quantization error decreases as alignment improves. However, as illustrated in Fig. 11(b), beyond this point, the trend reverses. We hypothesize that excessively large $\beta$ overemphasizes anomaly alignment at the expense of reconstruction fidelity, diminishing the influence of $\mathcal{L}_{\text{rec}}$. In this regime, the optimized rotations $\boldsymbol{R}$ achieve tighter alignment, but simultaneously distort the activation geometry. Such distortion reduces the representational capacity of the rotated features, hindering the quantizer's ability to preserve fine-grained information across timesteps, and ultimately increases quantization error. In other words, alignment alone is insufficient—accurate reconstruction is also essential for generalizing rotations across timesteps. Based on this trade-off, we adopt $\beta = 0.05$, which achieves the best empirical balance between anomaly alignment and reconstruction fidelity.

Table 7: The effect of varying population size $\mathcal{S}$ and iterations $N$ in the evolutionary search on quantization performance and search overhead.

| Population Size $\mathcal{S}$ | Iterations $N$ | FID ↓ | sFID ↓ | Search Time ↓ |
|---|---|---|---|---|
| 0 | 0 | 6.38 | 20.22 | 0 |
| | 10 | 6.35 | 20.19 | **0h 10min** |
| 20 | 20 | 6.31 | 20.14 | 0h 22min |
| | 50 | 6.27 | 20.04 | 0h 58min |
| | 10 | 6.32 | 20.07 | 0h 25min |
| 50 | 20 | 6.20 | 19.78 | 0h 54min |
| | 50 | 6.18 | 19.73 | 2h 18min |
| | 10 | 6.28 | 19.85 | 0h 51min |
| 100 | 20 | 6.18 | 19.70 | 1h 51min |
| | 50 | **6.16** | **19.65** | 4h 34min |

**Population size $\mathcal{S}$ and iterations $N$ in the evolutionary search.** As introduced in Sec. 3.3, Reordered Group Quantization (RGQ) employs an evolutionary search algorithm (Alg. 2) to determine the optimal channel permutation prior to group quantization. Two key hyperparameters govern this process: the population size $\mathcal{S}$ and the number of iterations $N$. We analyze their effects on both quantization performance and search efficiency, with results summarized in Tab. 7. Note that $\mathcal{S} = 0$ and $N = 0$ correspond to disabling RGQ entirely. The results show that increasing $\mathcal{S}$ and $N$ consistently improves performance (lower FID and sFID), as a larger search space and more refinement steps enable the discovery of higher-quality permutations through crossover, mutation, and selection. However, this improvement comes with a proportional increase in search time. For example, scaling from $\mathcal{S} = 20, N = 10$ to $\mathcal{S} = 100, N = 50$ reduces FID from 6.35 to 6.16, but inflates runtime from 10 minutes to over 4.5 hours. This indicates diminishing returns when both hyperparameters are excessively large: marginal quality gains no longer justify the steep computational overhead. Balancing efficiency and performance, we adopt $\mathcal{S} = 50$ and $N = 20$ in our main experiments, which yield substantial improvements while keeping search time within a practical budget.

## E LIMITATIONS

While TCA-DiT achieves strong quantization performance, it also has several limitations. First, both the rotation optimization and evolutionary search for channel reordering are performed in a layer-wise manner. As model size and depth increase, the cumulative computational overhead grows significantly, limiting scalability to very large diffusion transformers. Second, our study focuses exclusively on image generation tasks. Although image benchmarks provide a solid foundation for evaluation, the applicability of TCA-DiT to more complex modalities such as video generation remains unexplored. Extending our framework to video DiTs would not only demonstrate its generality but also enable more efficient generative modeling in spatiotemporal domains.

## F  LARGE LANGUAGE MODEL USAGE

We use ChatGPT (GPT-5, by OpenAI) as a writing assistant to polish the language of this paper. Specifically, it was employed to improve clarity, grammar, and style in certain sections, but it did not contribute to research ideation, technical content, experiments, or analysis. All scientific contributions, methods, and conclusions are solely the work of the authors. The authors take full responsibility for the accuracy and integrity of the paper's contents.

## G  VISUALIZATIONS

### G.1  ACTIVATION DISTRIBUTIONS COMPARISON

In Sec. 4.2, Fig. 7 illustrates the activation distributions of the `blocks.1.mlp.fc1` layer in the DiT-XL/2 model. Compared to the baselines, TCA-DiT produces a more uniform distribution, resulting in the lowest relative L1 quantization error. Here, we provide additional examples. Specifically, for the PixArt-$\alpha$ under W4A4 with 20 denoising steps, we generate an image using a random COCO prompt and compare activation distributions at timestep 19 ($t = 19$) across three layers: `blocks.2.attn1.to_q`, `blocks.2.attn1.to_out`, and `blocks.27.ff.net.0.proj`, as shown in Fig. 12. The results show that TCA-DiT consistently yields more uniform activation distributions than the baselines, thereby achieving the lowest quantization error. These findings further validate its effectiveness and robustness in low-bit quantization.

### G.2  VISUAL QUALITY SAMPLES

To further assess the effectiveness of TCA-DiT, we present qualitative comparisons under different quantization settings. For **DiT-XL/2**, we evaluate on ImageNet at both 256×256 and 512×512 resolutions with W4A8 and W4A4 quantization, using a CFG scale of 4.0 and 100 denoising steps. Representative samples are shown in Figs. 13 to 20, where Figs. 13 and 14 and Figs. 15 and 16 correspond to ImageNet 256×256 under W4A8 and W4A4 respectively, and Figs. 17 and 18 and Figs. 19 and 20 correspond to ImageNet 512×512 under the same settings. Similarly, for **PixArt-$\alpha$-512**, we provide results in Figs. 21 to 24 under W4A8 and W4A4, using a CFG scale of 4.5 and 20 denoising steps with randomly sampled COCO labels. In all cases, we compare TCA-DiT against baseline quantization methods as well as the Full-Precision (FP) model.

The visual comparisons yield several key observations:

- **Fidelity and semantics.** TCA-DiT consistently produces generations most similar to the FP model, preserving sharper details, richer semantics, and more accurate object rendering than competing approaches.

- **Robustness under aggressive quantization.** At W4A4, baseline methods such as PTQ4DiT, Q-DiT, and DiTAS exhibit severe degradation, often yielding collapsed or distorted images. In contrast, TCA-DiT maintains stable performance and high perceptual quality even in this challenging setting.

- **Generalization across models.** On PixArt-$\alpha$, TCA-DiT outperforms prior state-of-the-art methods such as ViDiT-Q, demonstrating stronger text–image alignment and semantic consistency, underscoring its applicability across the DiT family.

In summary, these visualizations corroborate the quantitative results: TCA-DiT not only narrows the gap between quantized and full-precision models but also establishes new performance baselines in terms of both visual quality and robustness. This highlights the effectiveness of anomaly channel alignment in stabilizing low-bit diffusion transformers.

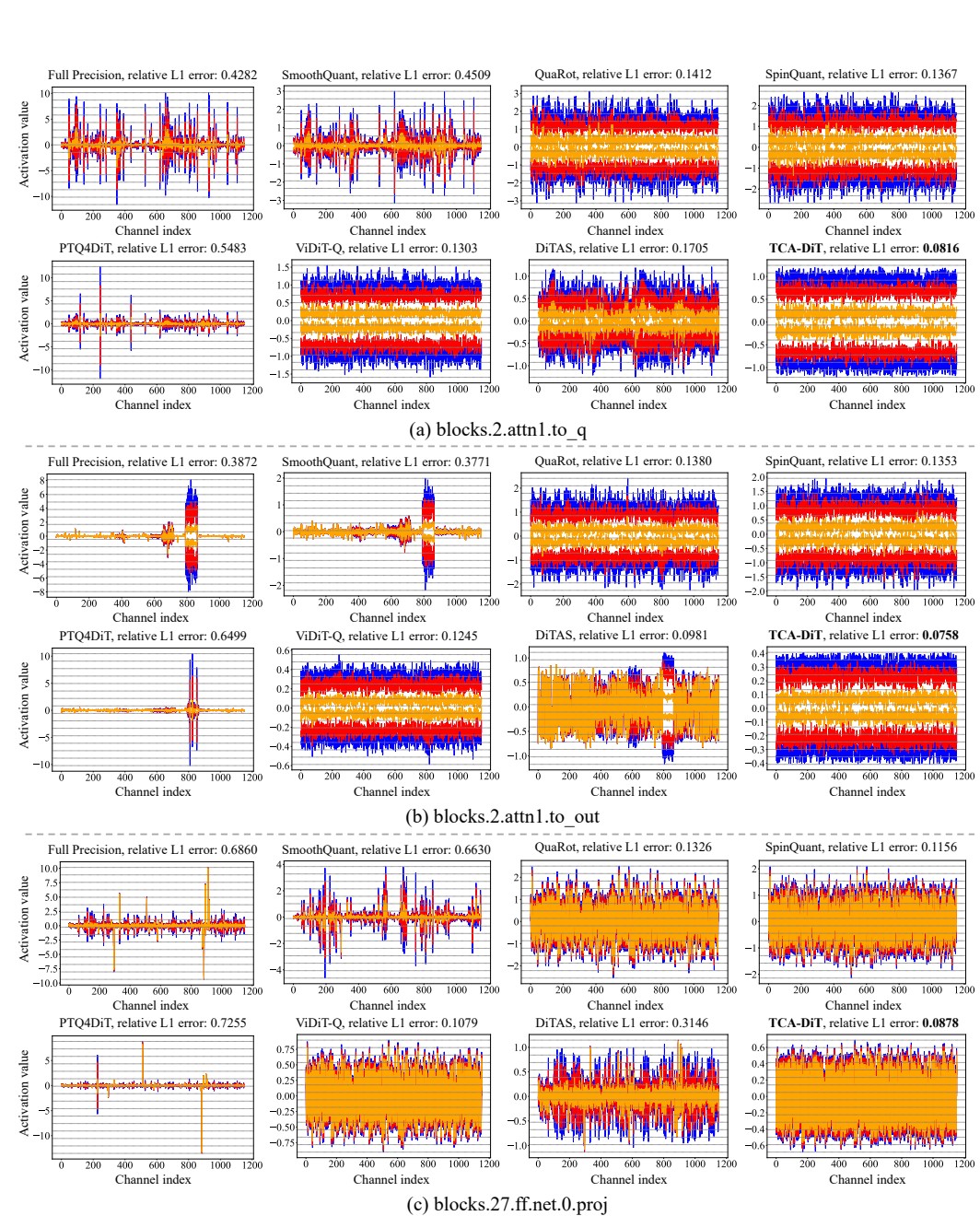

Figure 12: Activation distribution comparison between FP, baselines, and TCA-DiT on PixArt-$\alpha$.

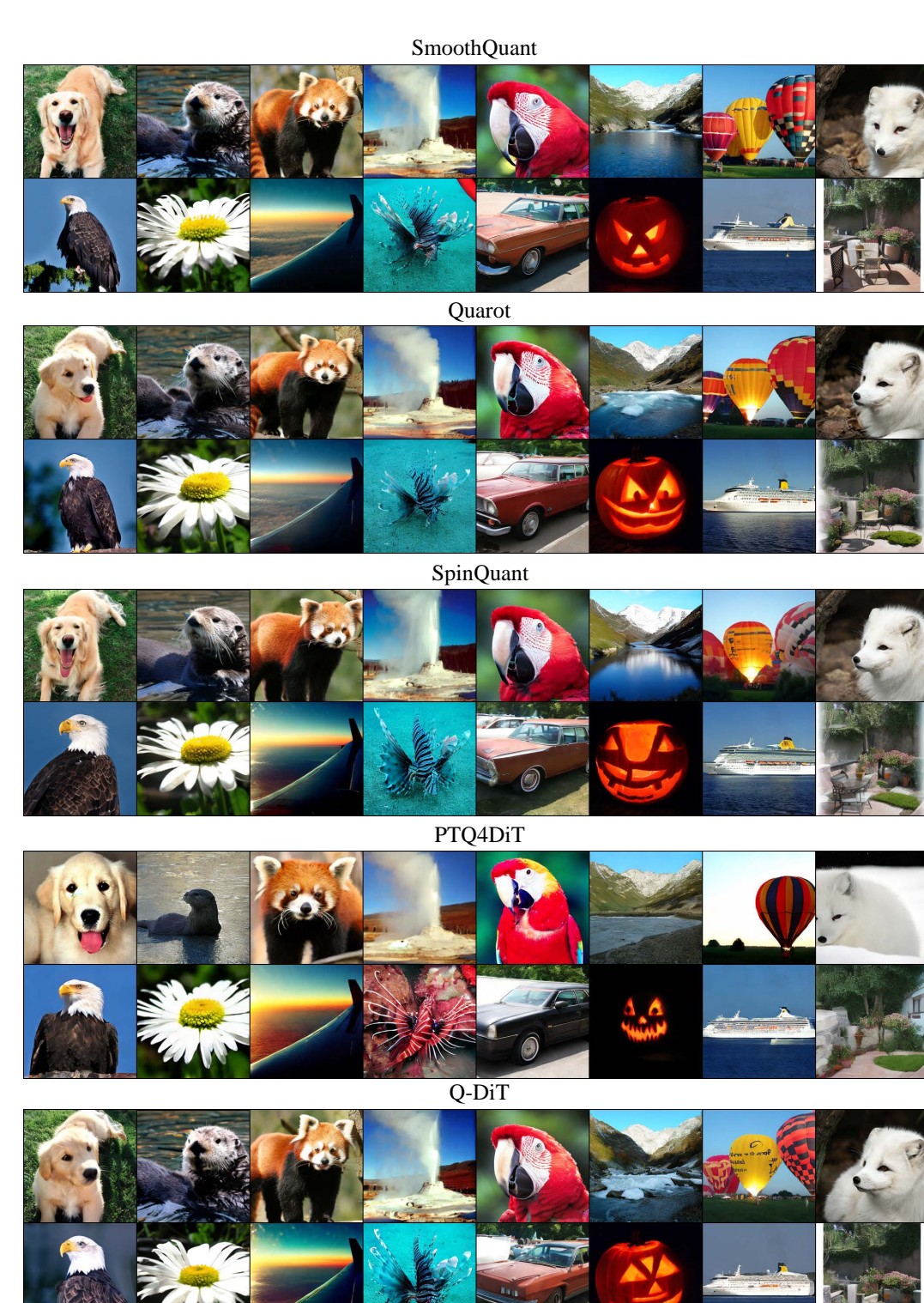

Figure 13: Random samples generated by different PTQ methods for DiT-XL/2 with **W4A8** quantization, on ImageNet 256×256 (cfg=4.0, denoising steps 100).

ViDiT-Q

DiTAS

**TCA-DiT**

Full Precision

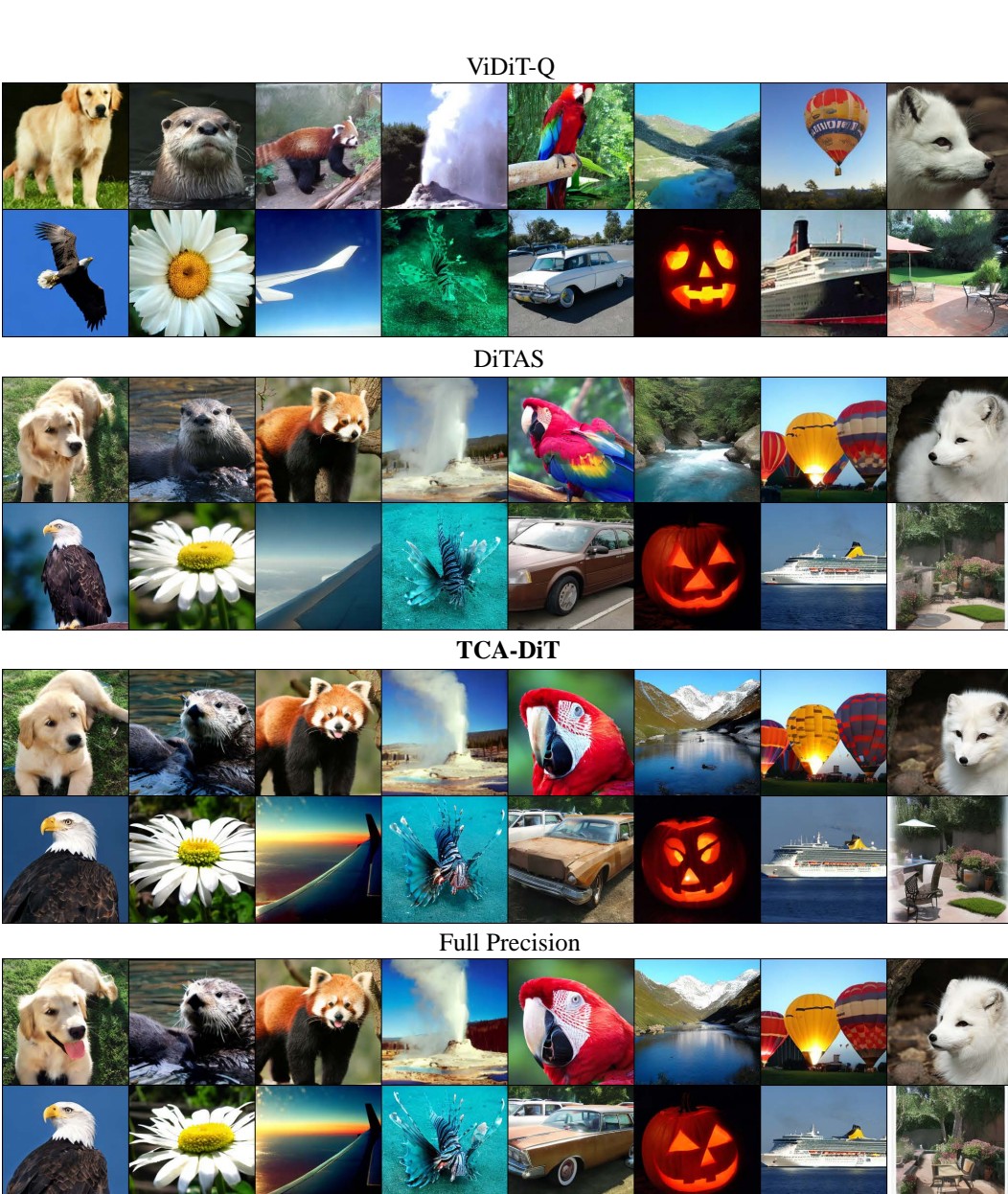

Figure 14: Random samples generated by different PTQ methods for DiT-XL/2 with **W4A8** quantization, on ImageNet 256×256 (cfg=4.0, denoising steps 100).

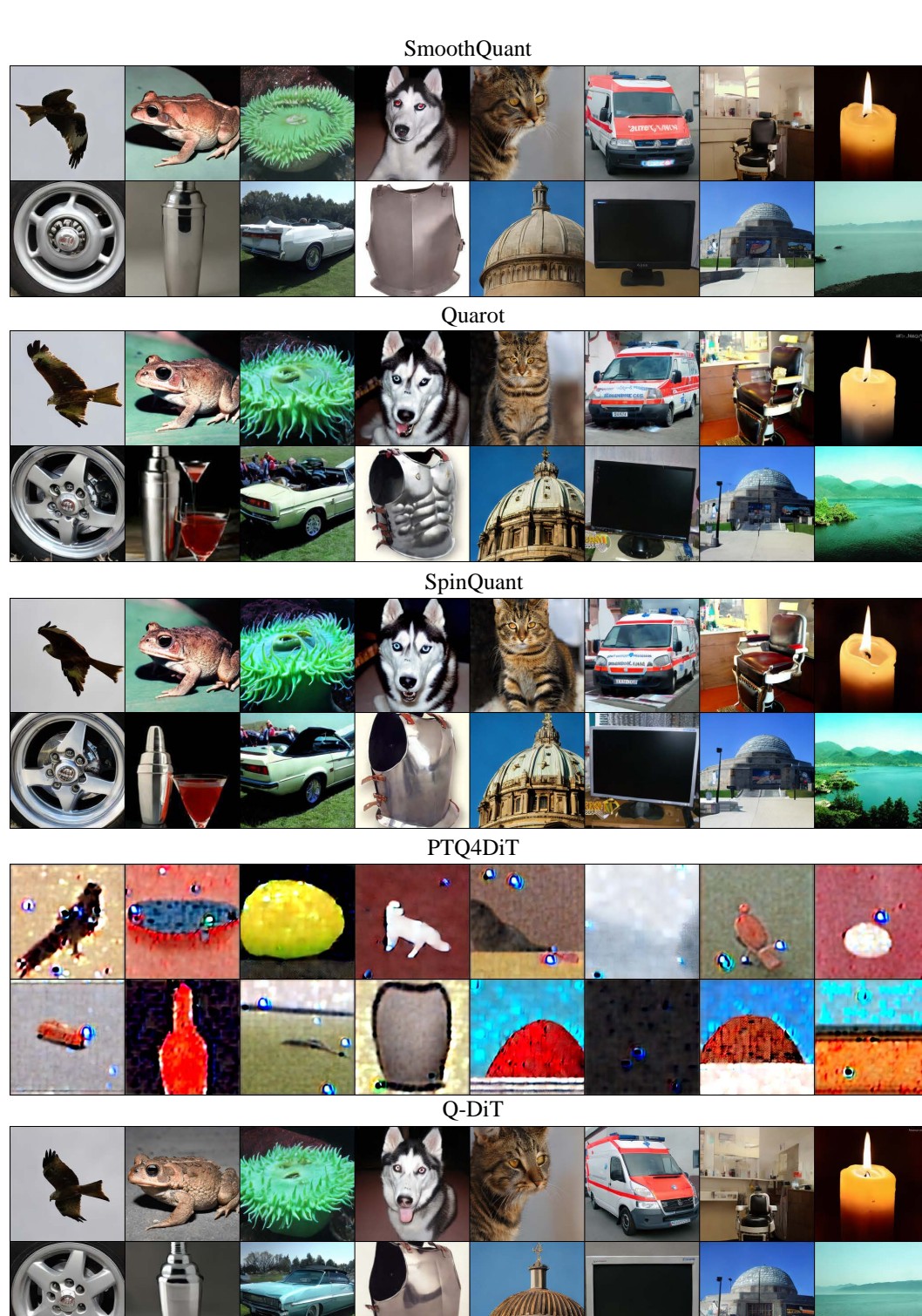

Figure 15: Random samples generated by different PTQ methods for DiT-XL/2 with **W4A4** quantization, on ImageNet 256×256 (cfg=4.0, denoising steps 100).

ViDiT-Q

DiTAS

**TCA-DiT**

Full Precision

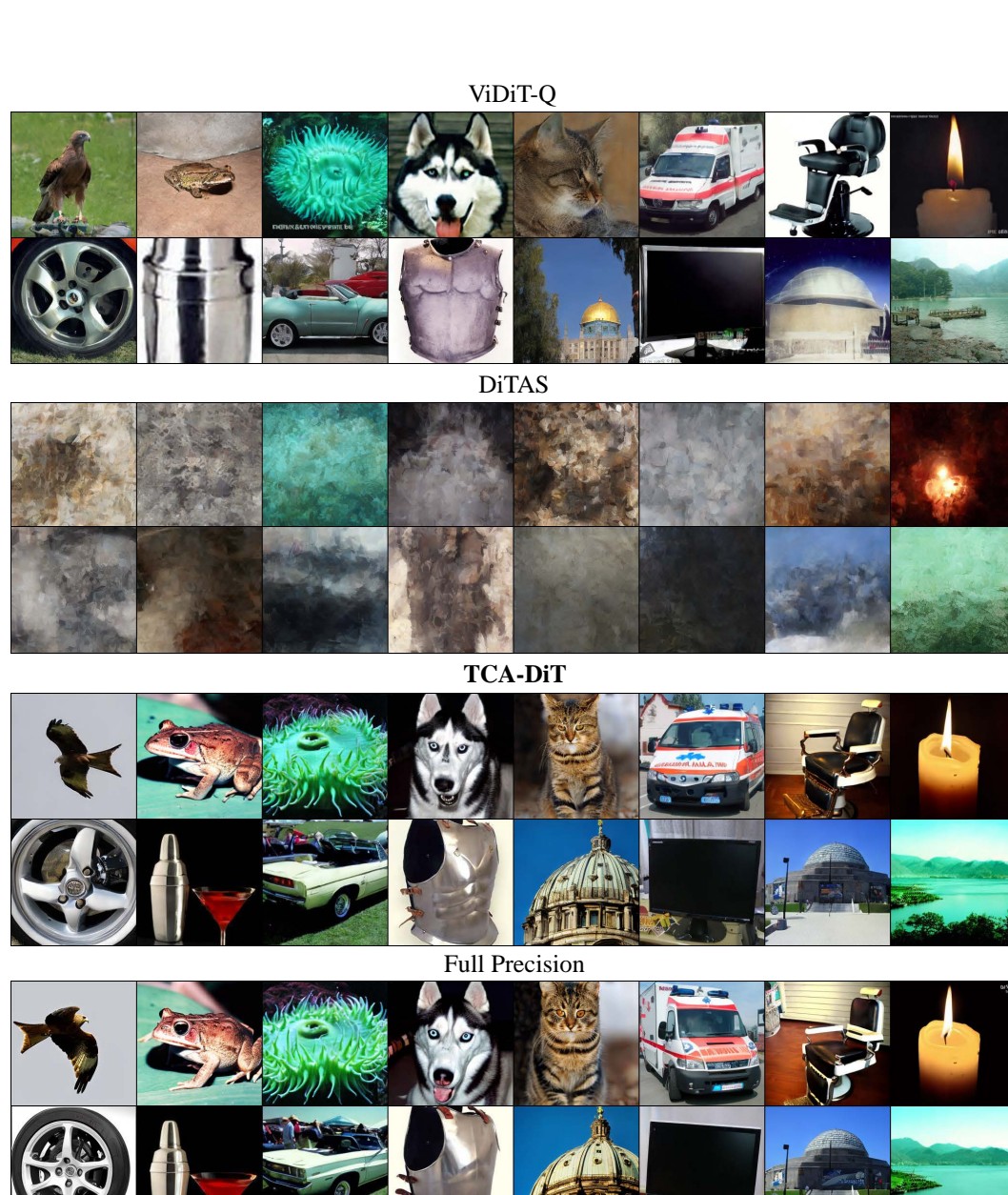

Figure 16: Random samples generated by different PTQ methods for DiT-XL/2 with **W4A4** quantization, on ImageNet 256×256 (cfg=4.0, denoising steps 100).

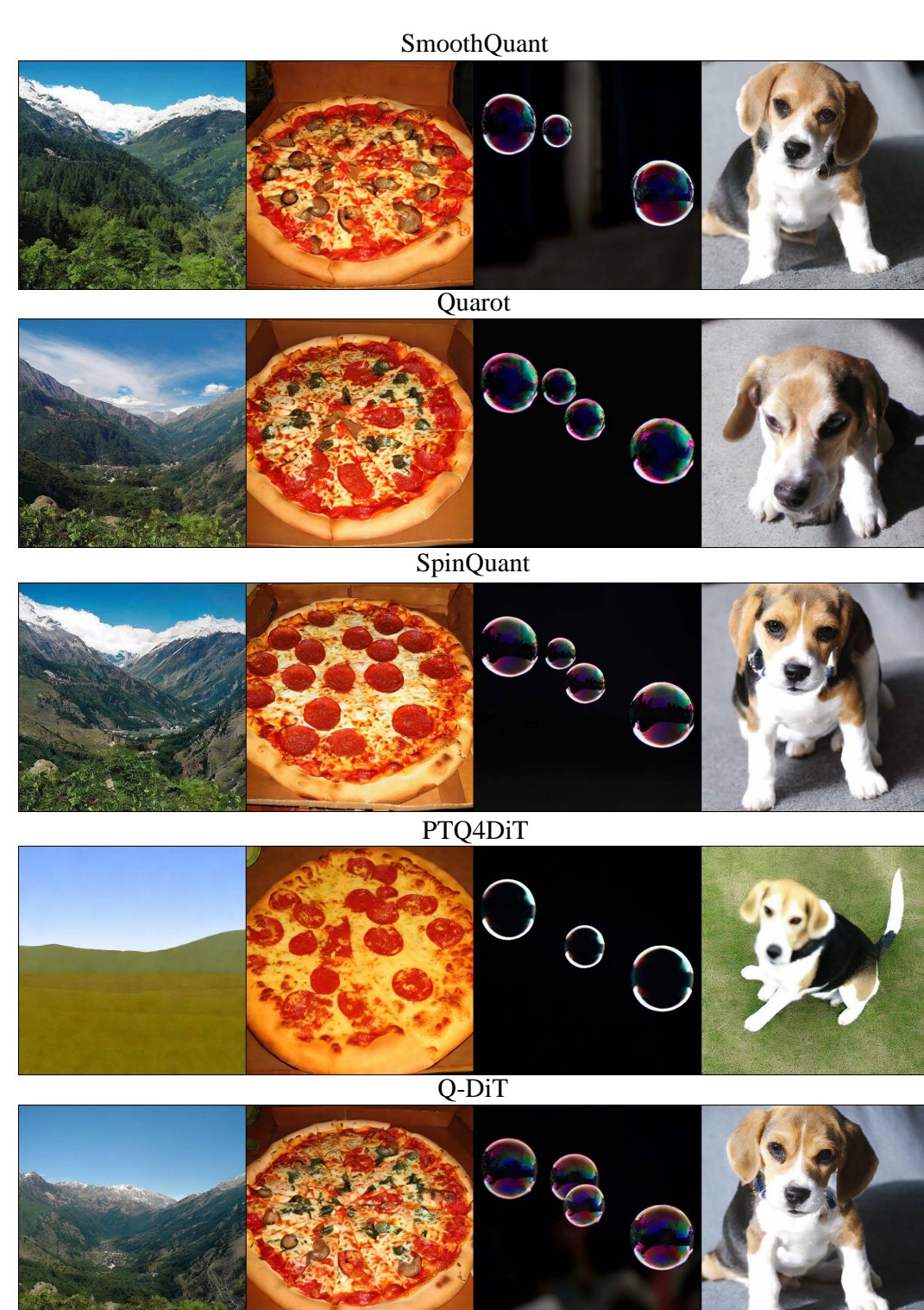

Figure 17: Random samples generated by different PTQ methods for DiT-XL/2 with **W4A8** quantization, on ImageNet 512×512 (cfg=4.0, denoising steps 100).

ViDiT-Q

DiTAS

**TCA-DiT**

Full Precision

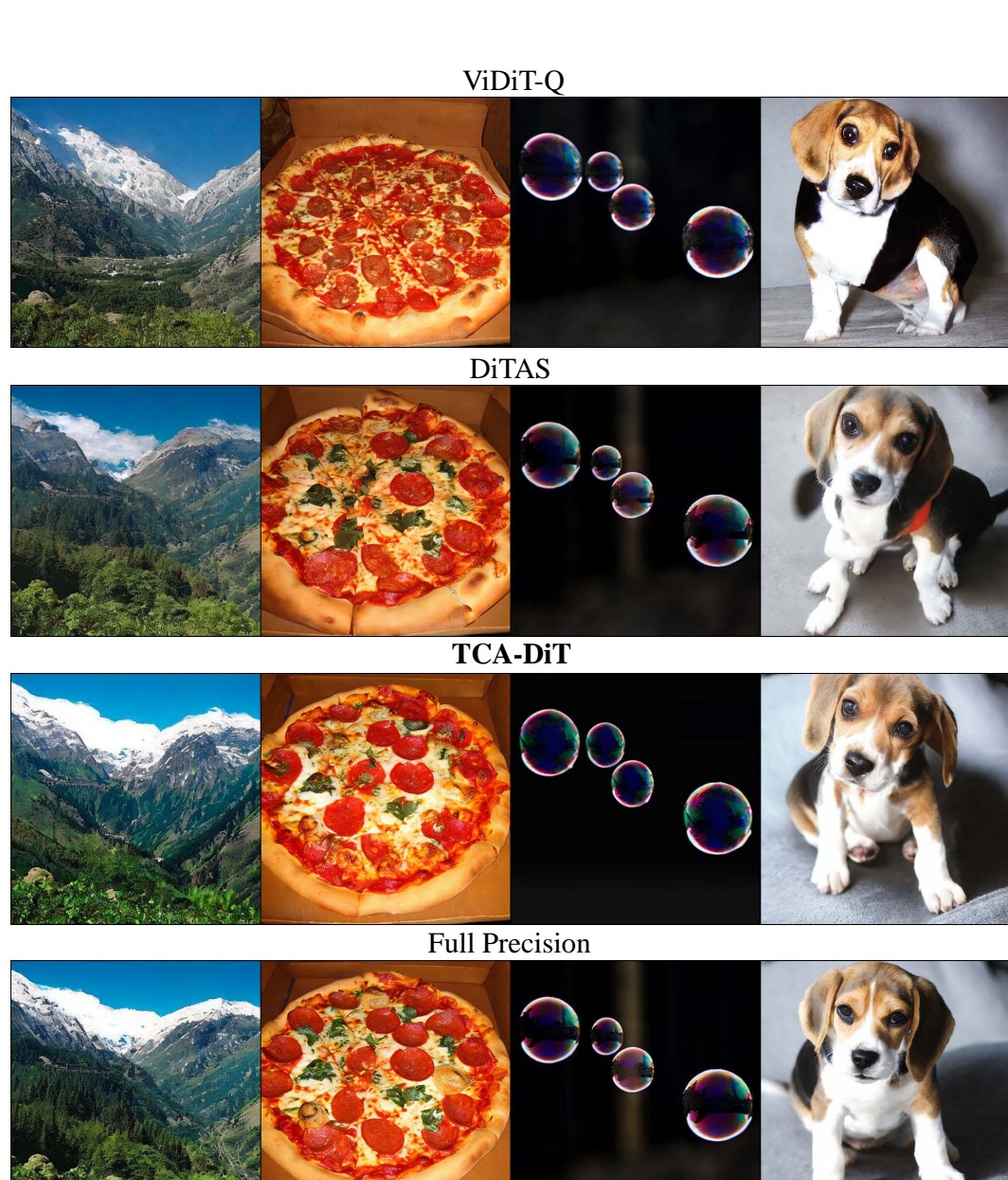

Figure 18: Random samples generated by different PTQ methods for DiT-XL/2 with **W4A8** quantization, on ImageNet 512×512 (cfg=4.0, denoising steps 100).

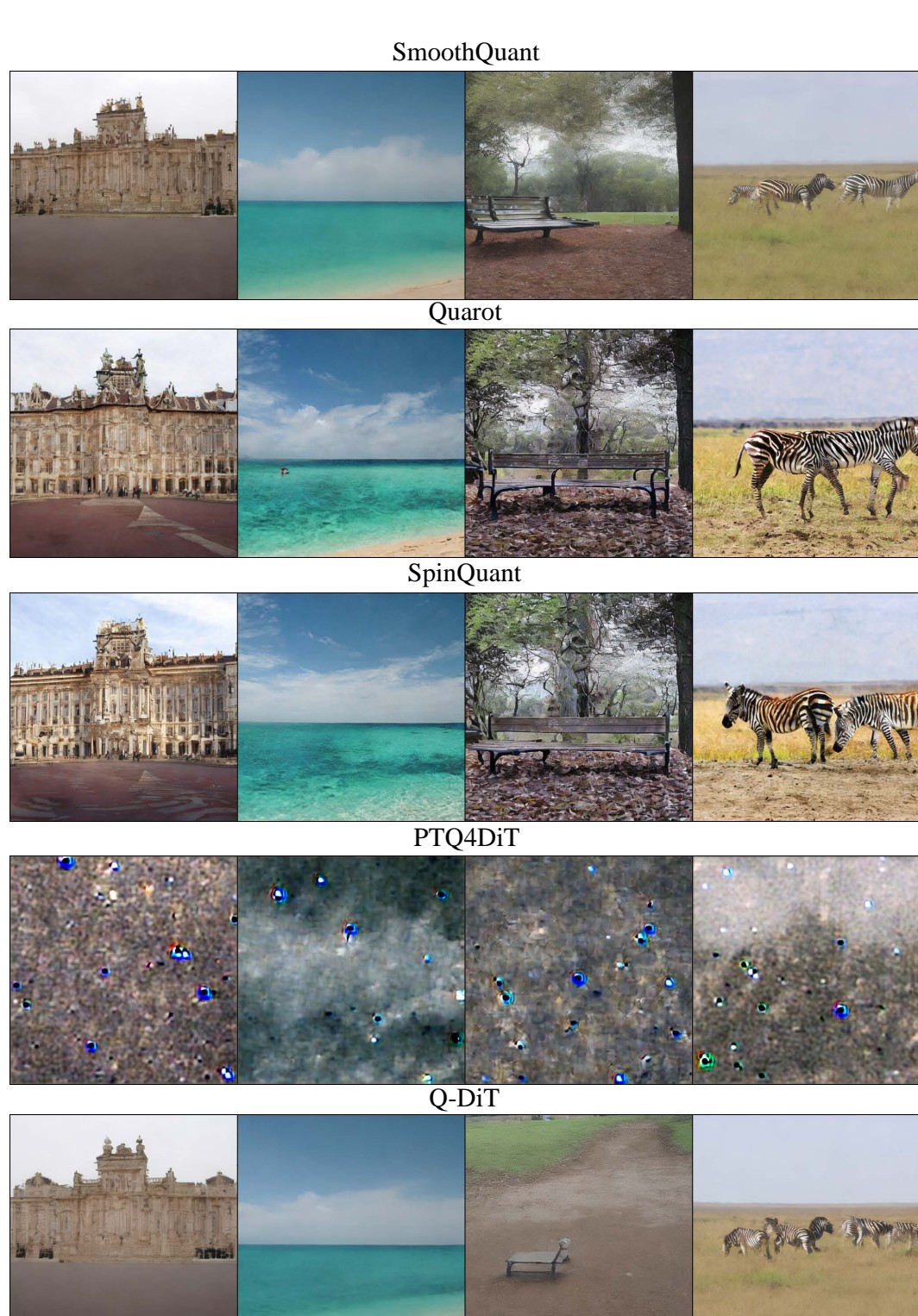

Figure 19: Random samples generated by different PTQ methods for DiT-XL/2 with **W4A4** quantization, on ImageNet 512×512 (cfg=4.0, denoising steps 100).

ViDiT-Q

DiTAS

**TCA-DiT**

Full Precision

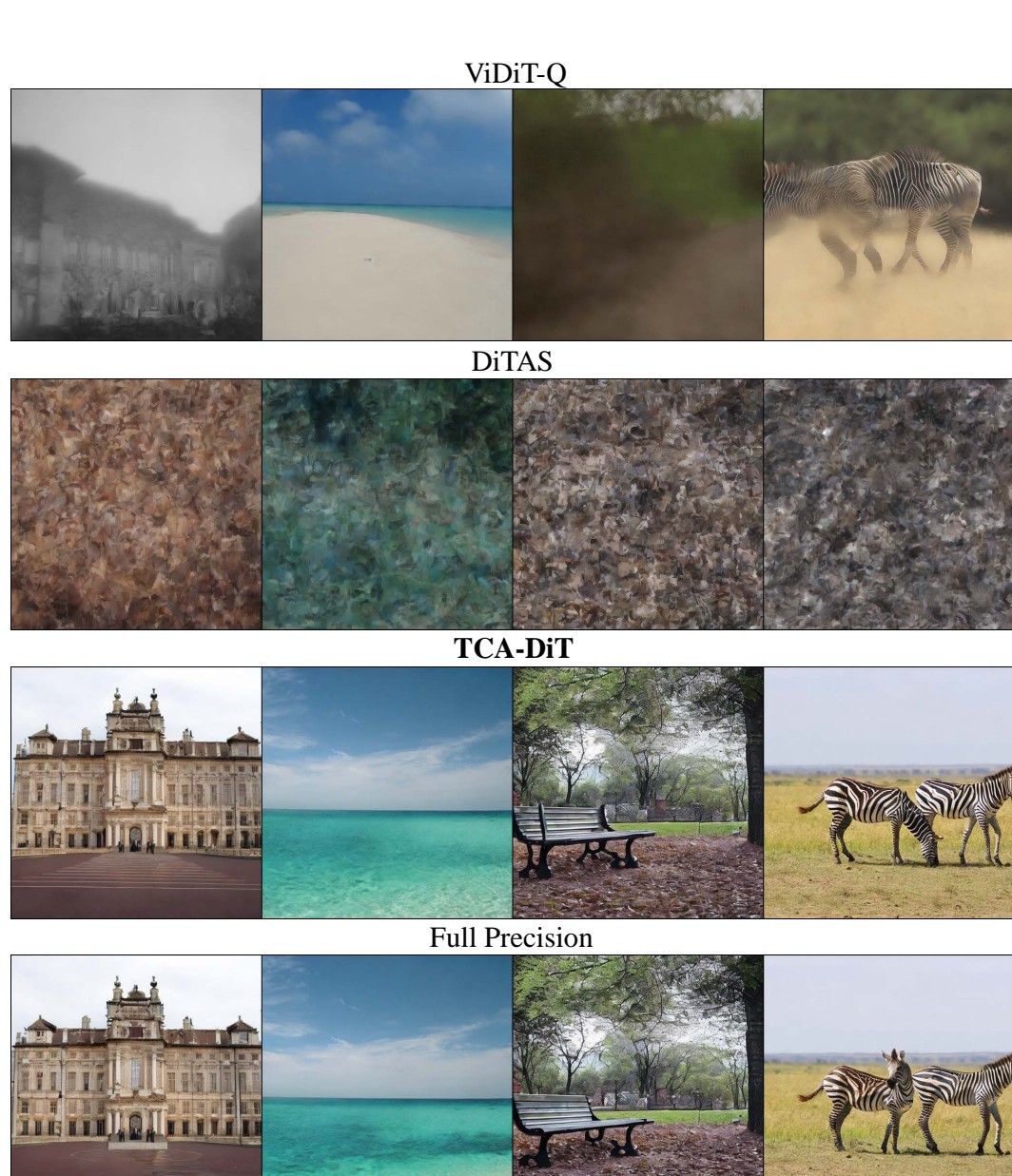

Figure 20: Random samples generated by different PTQ methods for DiT-XL/2 with **W4A4** quantization, on ImageNet 512×512 (cfg=4.0, denoising steps 100).

SmoothQuant

QuaRot

SpinQuant

PTQ4DiT

Q-DiT

Figure 21: Random samples generated by different PTQ methods for PixArt-$\alpha$-512 with **W4A8** quantization, on COCO captions (cfg=4.5, denoising steps 20).

ViDiT-Q

DiTAS

**TCA-DiT**

Full Precision

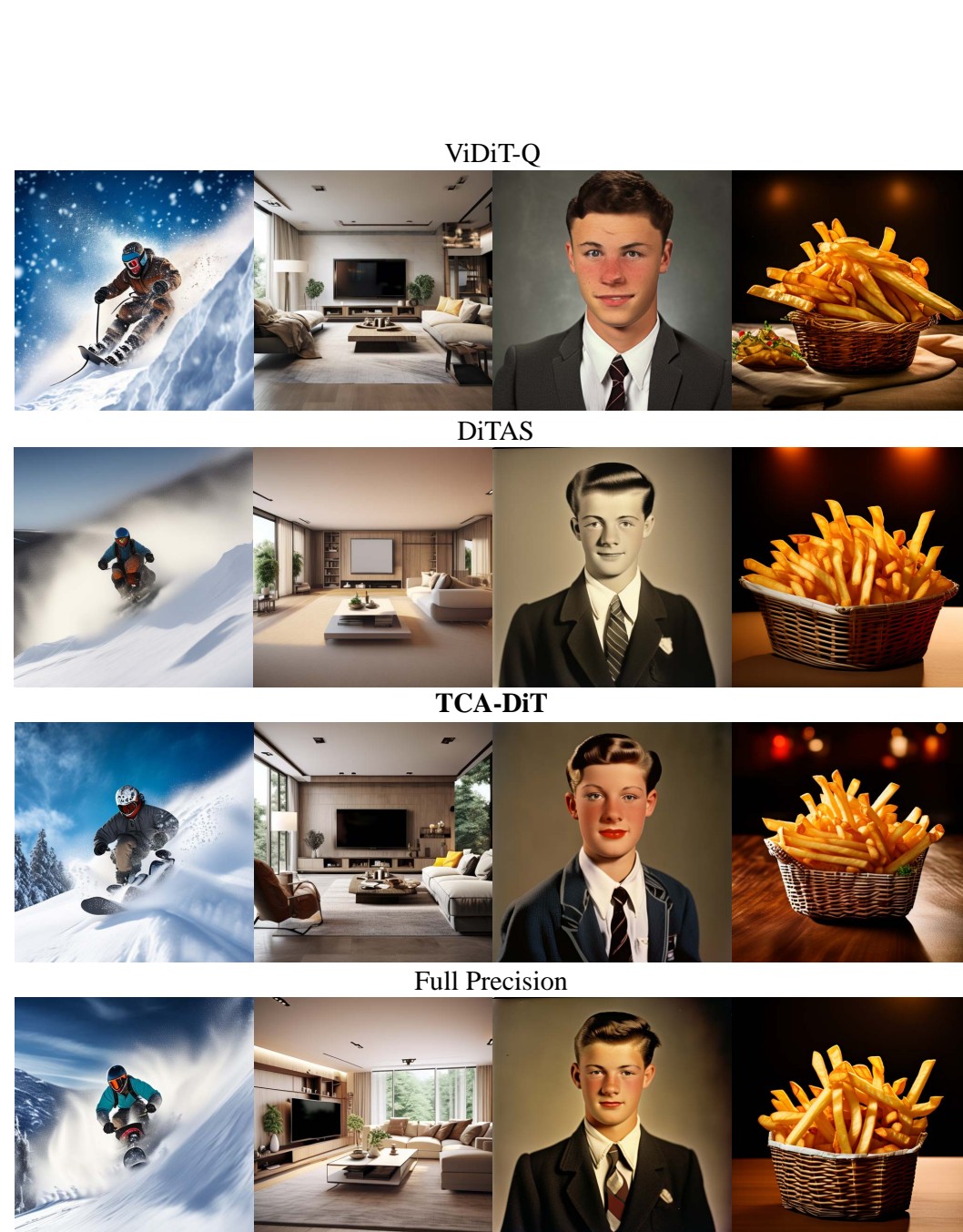

*A man riding down a snow covered slope in the snow.*   *A living room filled with furniture and a flat screen TV.*   *A person wearing a vest and tie in a yearbook photo*   *a basket of food on a table with fries*

Figure 22: Random samples generated by different PTQ methods for PixArt-$\alpha$-512 with **W4A8** quantization, on COCO captions (cfg=4.5, denoising steps 20).

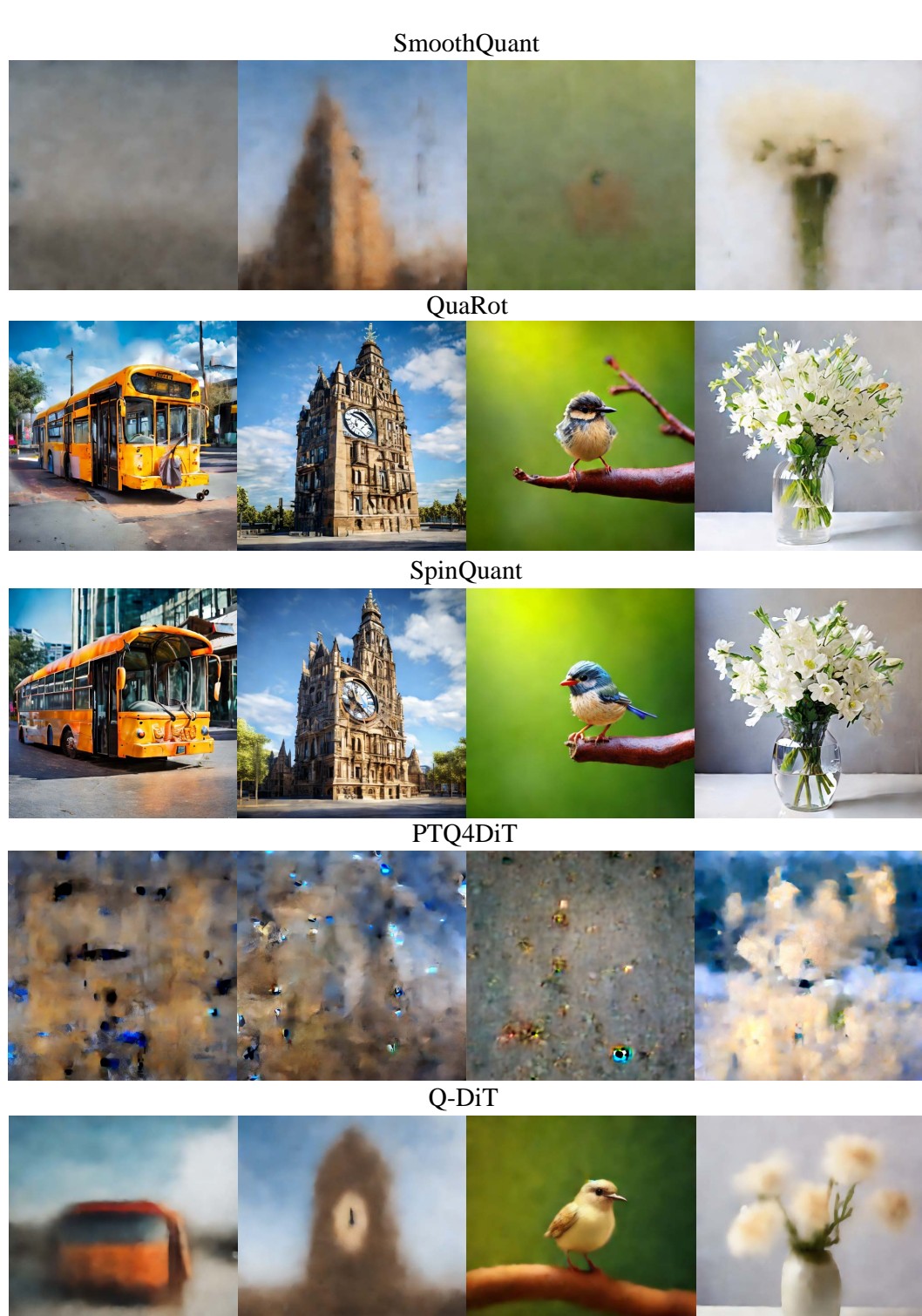

Figure 23: Random samples generated by different PTQ methods for PixArt-$\alpha$-512 with **W4A4** quantization, on COCO captions (cfg=4.5, denoising steps 20).

ViDiT-Q

DiTAS

**TCA-DiT**

Full Precision

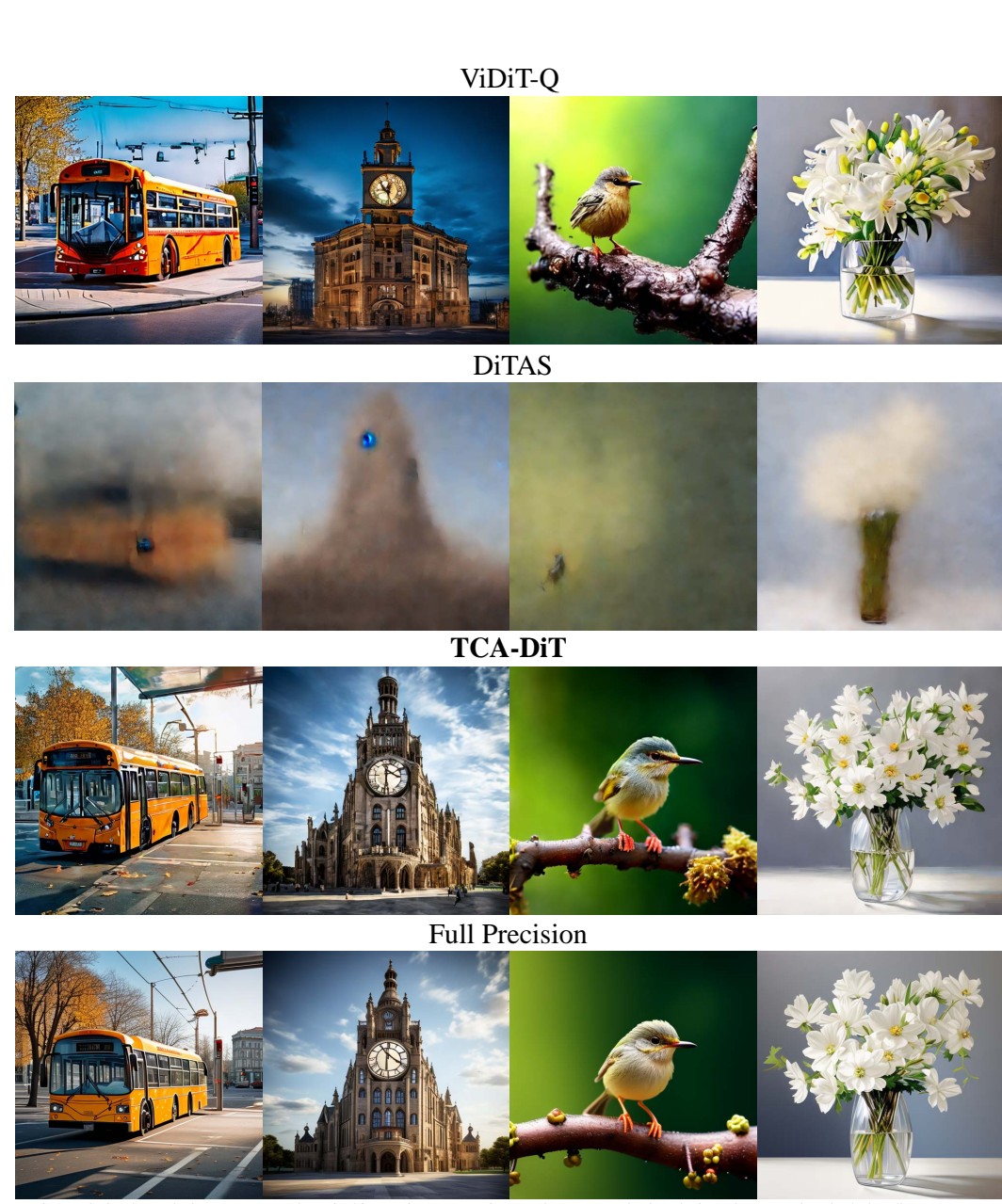

*A bus parked in front of a bus stop.*    *A large building with a tower and clock on top.*    *There is a tiny bird on the branch.*    *A bunch of white flowers is in a clear vase.*

Figure 24: Random samples generated by different PTQ methods for PixArt-$\alpha$-512 with **W4A4** quantization, on COCO captions (cfg=4.5, denoising steps 20).

