# OpenReview forum: "TCA-DiT: Quantizing Diffusion Transformers via Temporal Channel Alignment"
_ICLR.cc/2026/Conference — ICLR 2026 Conference Withdrawn Submission_

### Official Review · Reviewer_Yurq · 2025-10-15

**Soundness:** 2
**Presentation:** 2
**Contribution:** 2
**Rating:** 4
**Confidence:** 3

**Summary:**

The authors propose TCA-DiT, a method that suppresses outliers in diffusion transformers (DiTs) through applying transformations (rotation and rescaling and permutations). They introduce:
1. "Anomaly-aware Rotation Calibration", which learns (i) a rotation matrix through a layer reconstruction loss averaged over timesteps, and (ii) a scaling vector on top of the rotation, which aims to align the anomaly channels across time-steps.
2. "anomaly-guided timestep grouping", which aims to cluster timesteps with similar outlier distributions
3. "reordered group quantization", which aims to reduce quantization error post-transformation further by reordering channels such that block-wise quantization is improved.

**Strengths:**

I see true value in some of the methods ideas:
* quantization of diffusion models is an important topic, and small gains have big impact
* the temporal axis of outliers is hardly researched
* aligning the outlier channels across time steps makes a lot of sense to me for making transform and block quantization more effective

**Weaknesses:**

# Method

Despite being very familiar with this area, I find the method hard to follow and have many questions.

## Section 3.1

1. Where are the rotations applied, before all linear layers?
2. If so, authors claim they can be merged into previous operations. Is this always true?
3. How do static rotations (i.e. every time step uses the same rotations) provide a tool to align the outlier channels? This may relate to 1---I don't understand where the rotations are applied exactly in the network
3. The anomaly alignment loss (Eq. 3 and surrounding text) is difficult to follow as it's unclear where in the network we are even looking at, and thus what X refers to or why there is a double softmax.
4. Section 3.1, Stage 2, is literally SmoothQuant, but this is not cited or acknowledged. Do you agree?

## Section 3.2

It is unclear how relevant ATG is to the method's success.
1. The "ATG" contribution is needed for efficiency of Eq. 3, but what is really the gain? The acativations throughout the diffusion process still need to be acquired, so it's only the direct sum in Eq. 3 that is approximated. It's unclear whether this gives efficiency gains and whether there is a penalty in terms of accuracy.
2. More modern diffusion models usually use fewer time steps (e.g. as you write in the text-to-image results, just 20 steps), so it's less of an issue. Likely clustering is less effective as local correlation with fewer time steps will be lower
3. It is also unclear how expensive this clustering is---we still need to get all the activations and optimize (Eq. 6)
4. Also not clear how much better it is than just uniformly splitting the diffusion process into equally sized groups. The "risks overlooking critical transitions" [L269] are not clear, looking at Fig. 5a.

Overall, I would consider showing either that ATG is essential, or to remove it/make it an implementation detail and not part of the method.

## Section 3.3

1. This seems very much tangential to the previous contributions.
2. Why can the permutation (which is an orthogonal matrix) not be merged into the earlier rotation matrix?


## Language and readability

Not as important, but language doesn't help readability, because it is imprecise and ambiguous at times. A few examples:
* What does "anomaly-aware" or "anomaly-guided" do, that related work that addresses outliers (i.e. almost all quantization literature) does not do? Also, why choose the word "anomaly", if everyone else uses "outlier"?
* "residual" is used in many places (e.g. the ambiguous "residual anomaly channels") but it is unclear what this refers to. Does it refer to the residual stream? Or is it meant as "remaining"?
* [L269] "naive uniform subsampling", do you mean, splits of $k$ consecutive steps?

Also, (minor) your citation style is wrong throughout (with a few exceptions). Use `\citep` typically, not `\cite` or `\citet`---see the ICLR style guide 🙂

# Results

Overall, results are quite exhaustive and convincing. A few points:
1. Figure 9: no baselines in benchmark. It would be useful to get insight into the overhead of the rotate/scale/permute.
2. The ablation would be more interesting on text-to-image generation which is the most important nowadays. I suspect that ATG doesn't help there because of the fewer time steps.
3. There are no standard deviations, so especially in the ablation, it is not clear at all whether the different improvements (e.g. FID down by 0.1x ) is significant
4. The baselines do not match the numbers from respective papers (ViDiT-Q and DiTAS). Can you explain why?

**Questions:**

See weaknesses

---

### Official Review · Reviewer_Ly4w · 2025-10-22

**Soundness:** 2
**Presentation:** 3
**Contribution:** 2
**Rating:** 4
**Confidence:** 4

**Summary:**

This paper introduces TCA-DiT: a method for Diffusion Transformer quantization based on the alignment of channel outliers across different diffusion time steps based on rotation and scaling factors learned on a calibration step. The paper motivates and describes the procedure to learn across-time-stamp alignment through the addition of an ad-hoc regularization term. TCA-DiT further introduces a permutation to improve group quantization results. The experimental section compares DCA-DiT against a wide range of methods in DiT Quantization literature, demonstrating the effectiveness on various popular metrics.

**Strengths:**

* The paper provides a novel method to tackle the issue of varying activation statistics across multiple timestamp without requiring time-dependent weight copies.

* The paper reports an extensive description of the method providing numerous baselines and ablation studies. TCA-DiT consistently outperforms the baselines according to the provided metrics.

* The paper includes an assessment of the speedup, model, and inference memory usage.

**Weaknesses:**

* The quantization settings used for the experiment in Figure 6 are not clear from the main text since some models use per-group quantization, while other per-channel/per-token quantization. The details are specified in Appendix C2, but comparing the method using different quantization settings is not entirely fair nor clear.


* The main text includes little to no details on where the transformations are applied and how they are fused into the architecture. One side can be clearly fused into the weight matrix, but what happens to the other side is not clarified. Considering that the proposed method involves using arbitrary rotation matrices, this detail has quite a substantial impact on the model runtime.


    * The schema in the top right corner of Figure 3 is quite confusing since it looks like the Rotation and Scaling operation are applied to the output of the matmul, and the transpose is not explicit. How can $R$ and $S^{-1}$ be fused into other operations is also unclear.


* The paper reports that “A pilot study on the blocks.26.attn.proj layer of DiT-XL/2 further reveals that default channel ordering is suboptimal: random permutations can reduce reconstruction error and strongly correlate with FID (Fig. 5b)”.  Please clarify that this statement holds when considering per-group quantization. Overall the group quantization permutation seems perpendicular to the rest of the method.

**Questions:**

1. How does QuaRot + RGQ perform? The addition of this result in Figure 8 could help better disentangle the effects of the proposed improvements.

2. Figure 3 seems to imply that the rotation and scaling operation can be fused into other operations. Can the authors further clarify where this transformation can be fused? Linear layers in recent DiT architectures, such as PixArt-Alpha are preceded by a time-dependent normalization or non-linearities, which make these operations non-fusable.

3. How is the overhead in Figure 9 computed? Does that include the overhead of the un-fused rotations and scaling?  Previous work seems to report larger overheads even with optimized Fast Walsh-Hadamard Transforms, which are substantially faster than a dense matrix multiplication. Where are the transformations applied within each architecture? Some details are described in the appendix, but crucial aspects should be included in the main text to put the results from Figure 9 in context.

---

### Official Review · Reviewer_cf2t · 2025-10-24

**Soundness:** 3
**Presentation:** 4
**Contribution:** 3
**Rating:** 6
**Confidence:** 3

**Summary:**

This paper proposes TCA-DiT, a training-free PTQ pipeline for diffusion transformers (DiTs) that explicitly addresses **timestep-varying activation outliers**. It has three parts: ARC that learns an orthogonal rotation jointly with a reconstruction loss and a KL-based *temporal anomaly alignment loss*, then apply per-channel scaling, ATG that clusters timesteps to pick a compact calibration subset, and RGQ that evolutionary searchs to reorder channels before group-wise quantization. The method reports strong empirical W4A4 results on DiT-XL/2 (ImageNet) and PixArt-$\alpha$  with claimed 3.5$\times$ speedup and 3.8$\times$ memory reduction after fusing transforms into adjacent ops.

**Strengths:**

* Beyond generic time-aware DiT PTQ, this method proposed a novel and solid approach with a clear operational recipe:  rotate $\to$ align $\to$ scale, following experiment justification for PTQ of DiT models over diverse baselins. The pipeline is well-specified (Alg. 3).
* The experiment result is empirically strong at low bits: W4A8/W4A4 comparisons vs. SmoothQuant/QuaRot/SpinQuant/PTQ4DiT/ViDiT-Q/DiTAS show consistent wins; qualitative samples at W4A4 look markedly better.

**Weaknesses:**

* **Missing baseline.** SVDQuant (ICLR 25') is a *training-free W4A4* approach that **migrates outliers into a low-rank branch** and fuses it with INT kernels; it reports strong DiT results (PixArt/FLUX) and practical speedups. Not comparing against SVDQuant leaves the empirical case incomplete, especially at low bits where SVDQuant is competitive.
* **Missing ARC ablations.** The mid-timestep anchor is motivated (for balanceness) but **not ablated**; please add quantitative comparisons to alternative anchors (early/late/learned) and expanded robustness for the softmax-weighted aggregator ($SW_\gamma$) beyond the current illustrative figures.

**Questions:**

1. Is there any memory overhead induced with the ARC?
2. Does the learned rotation in ARC transfer across seeds/prompts/datasets?
3. Why is the LPIPS metric evaluated and presented in fig. 1 but missing in the main experiment? Does LPIPS (in comparison to fp32 generation) also consistently outperform the baseline methods (experiment expected)?

---

### Official Review · Reviewer_jhBb · 2025-10-31

**Soundness:** 3
**Presentation:** 2
**Contribution:** 3
**Rating:** 4
**Confidence:** 2

**Summary:**

This paper proposes a new post-training quantization method for diffusion transformers. Specifically, the approach aligns channel outliers across time steps to facilitate a more effective rotation matrix and further reorders channel indices to improve the quality of quantization grouping. Experimental results demonstrate notable improvements in generation quality compared to existing baselines.

**Strengths:**

1. The idea of aligning channel anomalies across time steps is interesting and addresses an often overlooked issue in prior quantization works, which may otherwise lead to imbalanced quantization errors during the diffusion process.

2. The empirical results demonstrate that the proposed method achieves consistent improvements over state-of-the-art baselines. Additionally, the paper provides validation of practical speed-up on real hardware, which increases the credibility and relevance of the approach.

3. The paper is clearly written, well-organized, and easy to follow.

**Weaknesses:**

1. The post-training quantization procedure appears computationally expensive. For example, quantizing PixArt-α-512 requires more than 11 hours of processing time. Although the proposed optimization objective leads to a more effective rotation matrix, the added complexity significantly increases cost compared to methods like SpinQuant, which completes quantization in fewer than 3 hours. I understand that the performance gains may justify some trade-off, but the high overhead represents a practical limitation.

2. The performance of ViDiT-Q is inconsistent. In several cases (e.g., 100-step W4A4 results in Figure 6), it underperforms Q-DiT, which is not considered a strong baseline in prior literature [1]. Could the authors explain the reasons?

3. Table 1 does not include full-precision (FP) results for comparison, which makes it difficult to assess the relative degradation introduced by quantization. Additionally, the reported MSCOCO results are not aligned with values from existing works [1]. Could the authors explain the reasons?

4. Some figures have low readability due to small text (e.g., Figures 3 and 4). Improving the font size would enhance clarity and presentation quality.

[1] Zhao, Tianchen, et al. "Vidit-q: Efficient and accurate quantization of diffusion transformers for image and video generation." arXiv preprint arXiv:2406.02540 (2024).

**Questions:**

NA

---

### Note · Authors · 2025-11-20

I have read and agree with the venue's withdrawal policy on behalf of myself and my co-authors.